# Macrophage achieves self-protection against oxidative stress-induced ageing through the Mst-Nrf2 axis

Ping Wang[1,2], Jing Geng [1,2], Jiahui Gao[1], Hao Zhao[1], Junhong Li[1], Yiran Shi[1], Bingying Yang[1], Chen Xiao[1], Yueyue Linghu[1], Xiufeng Sun[1], Xin Chen[1], Lixin Hong[1], Funiu Qin[1], Xun Li[3], Jau-Song Yu[4], Han You[1], Zengqiang Yuan[5], Dawang Zhou [1,2], Randy L. Johnson[6] & Lanfen Chen [1,2]

Reactive oxygen species (ROS) production in phagocytes is a major defense mechanism against pathogens. However, the cellular self-protective mechanism against such potential damage from oxidative stress remains unclear. Here we show that the kinases Mst1 and Mst2 (Mst1/2) sense ROS and maintain cellular redox balance by modulating the stability of antioxidant transcription factor Nrf2. Site-specific ROS release recruits Mst1/2 from the cytosol to the phagosomal or mitochondrial membrane, with ROS subsequently activating Mst1/2 to phosphorylate kelch like ECH associated protein 1 (Keap1) and prevent Keap1 polymerization, thereby blocking Nrf2 ubiquitination and degradation to protect cells against oxidative damage. Treatment with the antioxidant N-acetylcysteine disrupts ROS-induced interaction of Mst1/2 with phagosomes or mitochondria, and thereby diminishes the Mst-Nrf2 signal. Consistently, loss of Mst1/2 results in increased oxidative injury, phagocyte ageing and death. Thus, our results identify the Mst-Nrf2 axis as an important ROS-sensing and antioxidant mechanism during an antimicrobial response.

[1] State Key Laboratory of Cellular Stress Biology, Innovation Center for Cell Signaling Network, School of Life Sciences, Xiamen University, Xiamen, Fujian 361102, China. [2] Cancer Research Center of Xiamen University, Xiamen, Fujian 361102, China. [3] Medical College of Xiamen University, Xiamen, Fujian 361102, China. [4] Molecular Medicine Research Center, Chang Gung University and Liver Research Center, Chang Gung Memorial Hospital at Linkou, Taoyuan 33302, Taiwan. [5] The Brain Science Center, Beijing Institute of Basic Medical Sciences, Beijing 100850, China. [6] Department of Biochemistry and Molecular Biology, University of Texas, M.D. Anderson Cancer Center, Houston, TX 77030, USA. These authors contributed equally: Ping Wang, Jing Geng. Correspondence and requests for materials should be addressed to L.C. (email: chenlanfen@xmu.edu.cn)

The innate immune system plays an important role in preventing microbial invasion. However, its function is compromised with age[1]. How ageing impacts the self-renewal and plasticity of phagocytes remains unclear. Many theories of ageing have been proposed, including the free-radical and mitochondrial theories[2–4]. Both theories speculate that cumulative damage to proteins, lipids, and DNA by reactive oxygen species (ROS) is the major cause of ageing and antioxidant defense decreases with age. Oxidative damage affects mitochondrial DNA replication and transcription and results in decreased mitochondrial function, which in turn leads to enhanced ROS production and further oxidative damage to cells. ROS are also known to alter telomere structure and shorten its length to facilitate the ageing process[5]. However, macrophages engulf harmful microorganisms and destroy them in phagosomes, and these processes depend mainly on the production of large amounts of phagosomal and mitochondrial ROS[6–9]. Thus, the dedicated balance between the generation and elimination of ROS is essential to suppress excess ROS and thus attenuate ROS-induced damage and the ageing process in macrophages. How macrophages sense intracellular ROS levels and achieve the precise coordination of ROS generation and scavenging is still unclear. A more detailed understanding of the molecular mechanisms underlying the phagocyte ageing process should enable the development of strategies to overcome age-related antimicrobial defects and provide improved disease control and prevention for the elderly.

A previous study showed that knockdown of *Caenorhabditis elegans* CST-1, the orthologue of the Hippo kinase from *Drosophila*, accelerates ageing and shortens life span[10–12]. The kinases Mst1 and Mst2 (Mst1/2) are the mammalian Hippo homologs; these kinases inhibit cell proliferation and promote apoptosis during tissue development and regeneration by inhibiting their downstream effectors Yap and Taz through a kinase cascade comprising the scaffolding proteins WW45, RASSF, and Mob1 and the kinases Lats1/2 and Ndr1/2[13–21]. Although the antiproliferative role of Mst1/2 has been extensively studied, it is less well appreciated that Mst1 deficiency in both humans and mice results in a complex combined immunodeficiency syndrome characterized by recurrent bacterial and viral infections, lymphopenia and variable neutropenia[22–32]. We recently reported that toll-like receptor 4 (TLR4) signaling activates Mst1/2 kinases to augment huge amount of phagosomal and mitochondrial ROS generation through enhancing the juxtaposition of mitochondria–phagosome in phagocytes during an antimicrobial response[33]. In additions, hydrogen peroxide is commonly used to activate kinases Mst1/2 in vitro[14,19,20,34,35]. These data support the interplay between ROS and kinases Mst1/2.

Here, we demonstrate that oxidants generated endogenously by phagosomes or mitochondria recruit Mst1/2 from the cytosol to the phagosomal or mitochondrial membrane and markedly activated these kinases. We further show that Mst1/2 function as critical ROS sensors and attenuators through stabilizing nuclear factor (erythroid-derived 2)-like 2 (Nrf2, also known as NFE2L2), a major antioxidant transcription factor. Consistently, loss of Mst1/2 results in dramatically increased basal cellular ROS levels in macrophages. Thus, Mst1/2 may be key coordinators of ROS generation and scavenging in phagocytes. Loss of the Mst-Nrf2 signaling for maintaining redox homeostasis in macrophages may be partially responsible for the recurrent infections occurred in both Mst1-deficient humans and mice.

## Results

### Increased oxidative stress in Mst1/2-null phagocytes.
ROS are both important signals for maintaining the normal physiological function of cells and toxic species that cause cellular injury. We have previously shown that Mst1/2 signaling is required for the induction of ROS and bactericidal activity in phagocytes upon bacterial infection[33]. However, we surprisingly observed that bone marrow-derived macrophages (BMDMs) from $Mst1^{fl/fl}Mst2^{fl/fl}$ $Lyz$-Cre (DKO) mice exhibited higher basal levels of ROS than those from $Mst1^{fl/fl}Mst2^{fl/fl}$ wild-type (WT) animals (Fig. 1a). Further analysis suggested that loss of Mst1/2 can lead to oxidative stress in phagocytes. We observed that the levels of protein carbonylation and phosphorylated (p)-H2A.X, biomarkers of oxidative stress and DNA damage, respectively, were increased in DKO BMDMs compared to WT BMDMs, and N-acetylcysteine (NAC), a ROS scavenger, significantly downregulated the levels of protein carbonylation and p-H2A.X in DKO BMDMs (Fig. 1b–d). Furthermore, increased apoptotic events associated with enhanced cleavage of PARPγ and caspase 3 were found in DKO macrophages compared to WT BMDMs, while NAC treatment greatly reduced DKO BMDMs death (Fig. 1c, e, f). Consistently, DKO mice exhibited substantially increased apoptotic events and a decreased frequency of F4/80$^+$CD11b$^+$ macrophages in the bone marrow and spleen compared to their WT counterparts, and NAC treatment significantly reduced F4/80$^+$CD11b$^+$ cell apoptosis in the spleen of DKO animals (Fig. 1g, h; Supplementary Fig. 1a–c). In addition, the treatment of antimycin A or rotenone—two drugs that block the mitochondrial respiratory chain, lead to the generation of mitochondrial ROS and eventually increase cellular ROS, resulted in higher p-H2A.X levels and a higher cell death rate in Mst1/2 DKO macrophages than in WT cells (Fig. 1i–k; Supplementary 1d). These data indicate that Mst1/2 might play a role in protecting macrophages from oxidative stress.

### Premature ageing in Mst1/2-deficient macrophages.
Recently, Meng et al.[36] reported that the decay of redox-stress response capacity (RRC) is a substantive characteristic of aging. RRC refers to the ability of cells to respond to oxidative stress, specifically three major activities: the ability to generate ROS or RNS, the ability to regulate antioxidants, and the ability to degrade damaged proteins for maintaining cellular redox and protein homeostasis. We questioned whether loss of Mst1/2 causes premature ageing in macrophages because of the increased oxidative stress (Fig. 1), as well as, reported previously, the decreased respiratory burst and inability to clear bacteria[33].

It has been previously shown that activated peritoneal macrophages from aged mice expressed significantly lower levels of TLRs(1–9)[37]. Indeed, we confirmed that the expression levels of various TLRs (1–9) were much lower in aged WT macrophages (20 months) when compared with that of macrophages isolated from younger mice (2 or 12 months). In comparison of the TLRs in WT and DKO macrophages, we found that the expression levels of TLR1, 2, 5, 8, and 9 were not significantly different in 2-month old macrophages, but were significantly decreased in 12- and 20-month DKO macrophages. Whereas, the expression levels TLR3, 4, 6, and 7 were all much lower in DKO macrophages compared with that of WT macrophages at any given ages (Fig. 2a). In additions, we observed that the ability of macrophages to activate OVA-specific OT-II CD4$^+$ T cells as measured by the proliferation assay was gradually decreased with increasing age (Fig. 2b). Consistently, OT-II CD4$^+$ T cells exhibit much lower proliferation rate when incubated with 12- or 20-month DKO macrophages when compared with that of the same ages WT macrophages (Fig. 2b). Furthermore, transmission electron micrographs (TEMs) showed that more lipofuscin granules, a recognized hallmark of ageing[38], were present in peritoneal macrophages of DKO mice than in those of age-

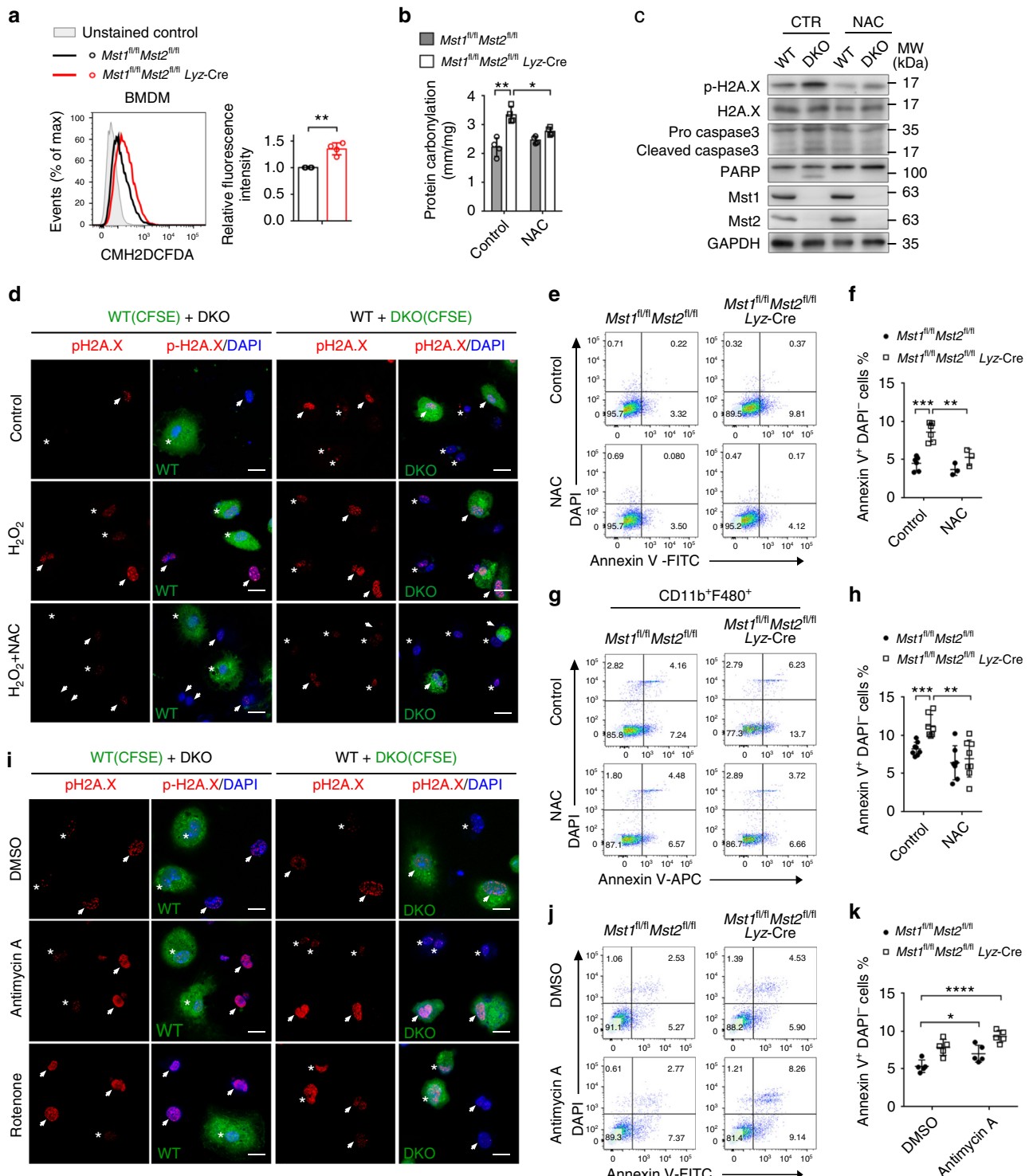

**Fig. 1** Increased ROS levels and apoptotic events in Mst1/2-null macrophages. **a** Flow cytometry of $Mst1^{fl/fl}Mst2^{fl/fl}$ (WT) and $Mst1^{fl/fl}Mst2^{fl/fl}$ Lyz-Cre (DKO) BMDMs stained for 30 min with the ROS dye CM-H2DCFDA. Quantification of the relative fluorescence intensity in the indicated cell samples is shown in the right panel. **b, c** Quantification of protein carbonylation levels (**b**) and immunoblot analysis of phosphorylated (p)-H2A.X (Ser139), H2A.X, pro- and cleaved-caspase 3, PARP, Mst1, Mst2, and GAPDH (**c**) in WT and DKO BMDMs pretreated with or without NAC. **d** Fluorescence microscopy of p-H2A.X staining (red) and DAPI− stained nuclei (blue) in WT BMDMs (left) or DKO DMDMs (right) labeled with CFSE, mixed with unlabeled DKO BMDMs (arrows) or WT BMDMs (stars), and treated with $H_2O_2$ and NAC as indicated. Scale bars, 20 μm. **e, f** Annexin V/DAPI staining (**e**) and quantification of Annexin V+DAPI− cells (**f**) among WT and DKO BMDMs pretreated with or without NAC. **g, h** Flow cytometry (**g**) and quantification (**h**) of the percentage of Annexin V+DAPI− cells among WT and DKO CD11b+F4/80+ cells pretreated with or without NAC. **i** Fluorescence microscopy of p-H2A.X staining (red) and DAPI-stained nuclei (blue) in WT BMDMs (left) or DKO DMDMs (right) labeled with CFSE, mixed with unlabeled DKO BMDMs (arrows) or WT BMDMs (stars), and treated with DMSO, antimycin A or rotenone. Scale bars, 20 μm. **j, k** Annexin V/DAPI staining (**j**) and quantification of Annexin V+DAPI− cells (**k**) among WT and DKO BMDMs treated with DMSO or antimycin A. *$P < 0.05$, **$P < 0.01$, ***$P < 0.001$, ****$P < 0.0001$ compared with control (Student's t test). Data are from one experiment representative of three independent experiments with similar results (mean and s.d. of $n = 3$ (**f**), $n = 4$ (**a, b**), $n = 5$ (**k**) or $n = 8$ (**h**) biological replicates)

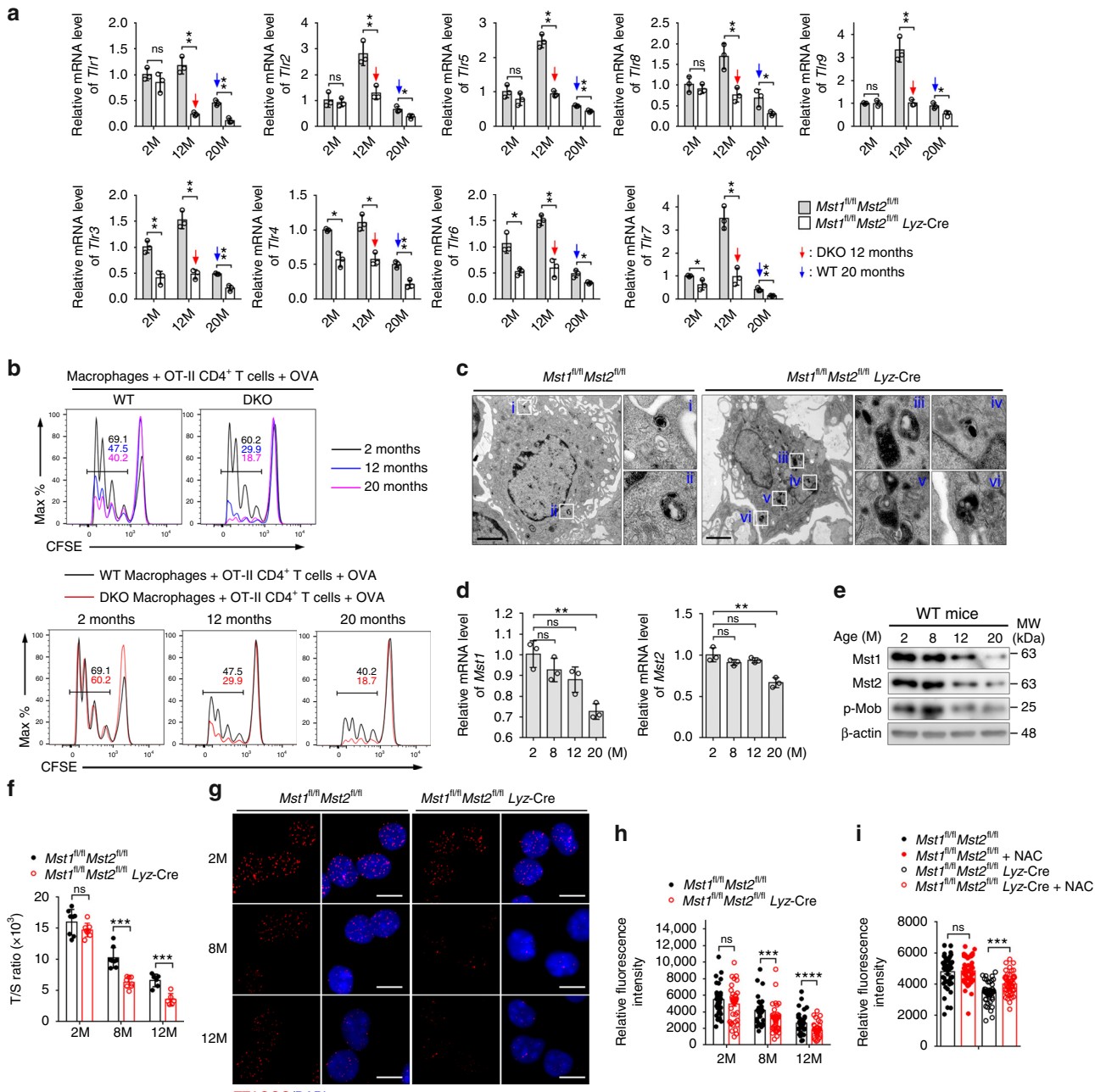

**Fig. 2** Loss of Mst1/2 promotes ageing in macrophages. **a** RT-qPCR analysis of various *Tlrs* genes on peritoneal macrophages isolated from *Mst1*fl/fl*Mst2*fl/fl (WT) or *Mst1*fl/fl*Mst2*fl/fl *Lyz*-Cre (DKO) mice with indicated age. **b** Proliferation assay of CFSE-labeled CD4+ OT-II T cells co-cultured with WT and DKO BMDMs in presence of OVA protein. **c** Representative transmission electron micrographs (TEMs) of lipofuscin granules within peritoneal macrophages isolated from 12-month-old WT and DKO mice. ×25 magnification of areas outlined in the main images as indicated. Scale bars, 2 μm. **d, e** RT-qPCR analysis of *Mst1* and *Mst2* (**d**), and immunoblot analysis of Mst1, Mst2 and p-Mob (**e**) in peritoneal macrophages isolated from WT mice with indicated age. **f–h** The relative telomere length (T/S ratio) (**f**), representative fluorescence microscopy images of telomere FISH analysis (red) and nuclei (blue) (**g**), and relative fluorescence intensity of telomere FISH (**h**) in peritoneal macrophages isolated from 2-, 8-, or 12-month-old WT and DKO mice. Scale bars, 10 μm. **i** Relative fluorescence intensities of telomere FISH in peritoneal macrophages isolated from WT and DKO mice with or without NAC supplementation in drinking water for 7 months. ns, not significant ($P > 0.05$); **$P < 0.01$, ***$P < 0.001$, ****$P < 0.0001$, compared as indicated (Student's *t* test). Data are from one experiment representative of three independent experiments with similar results (mean and s.d. of $n = 3$ (**a, d**, experimental replicates); $n = 5$ (**f**, biological replicates); mean and s.e.m. of $n = 50$ (**h, i**, experimental replicates))

matched WT mice at 12 months of age (Fig. 2c). Furthermore, the mRNA and protein levels of Mst1 and Mst2, as well as the p-Mob levels were dramatically reduced in aged macrophages (20 months) (Fig. 2d, e).

ROS-mediated accumulation of cellular damage results in increased telomere shortening and dysfunction, which are

associated with ageing, mortality and ageing-related diseases[5]. We quantified telomere length in peritoneal macrophages isolated from age-matched WT and DKO mice using real-time polymerase chain reaction (PCR) or fluorescence in situ hybridization (FISH), in which a fluorescently labeled probe complementary to the telomere sequence is hybridized and fluorescence intensity is

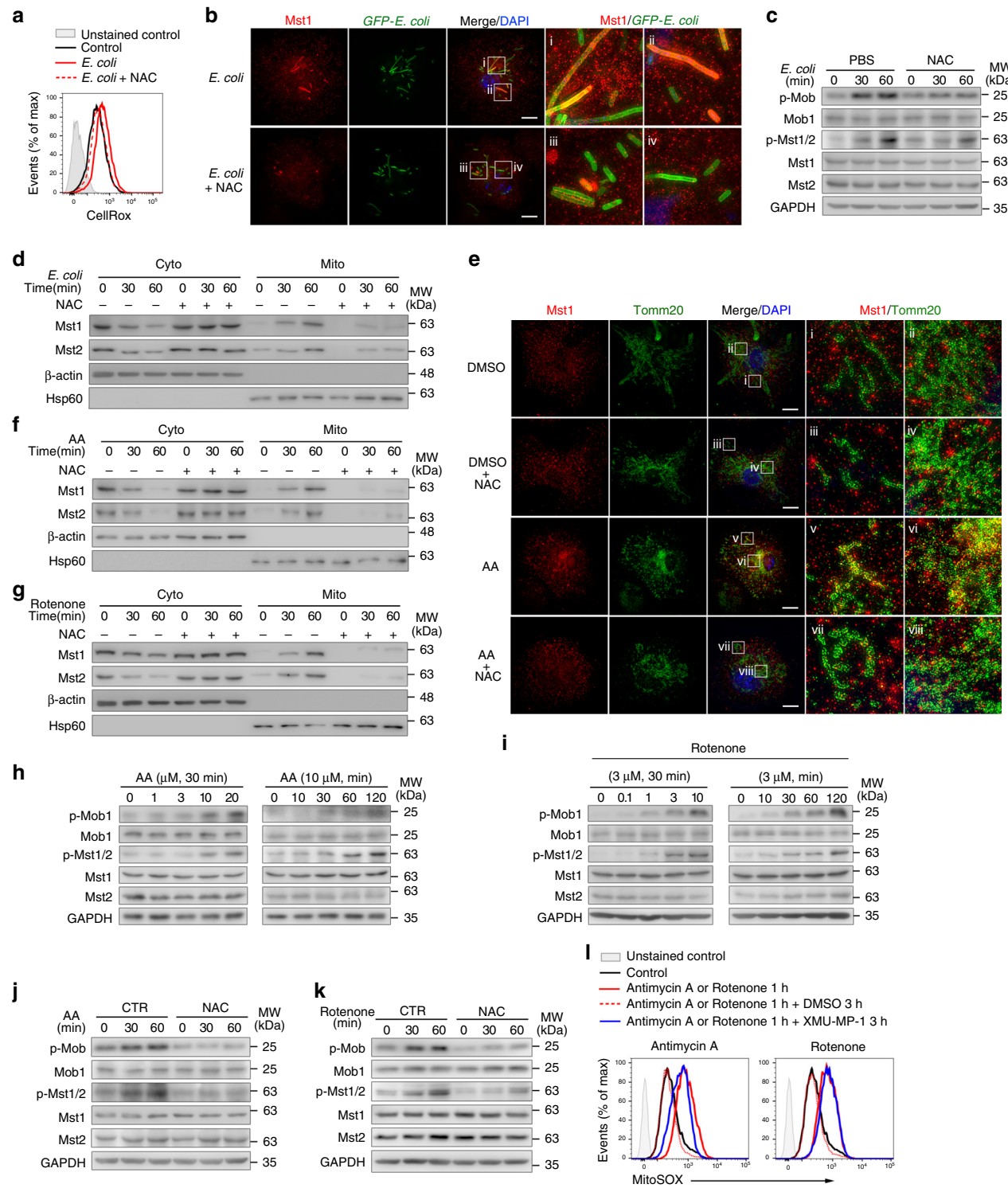

measured by immunofluorescence microscopy. DKO macrophages had shorter telomeres (Fig. 2f) and fewer telomere foci (Fig. 2g, h) than WT cells at 8 and 12 months of age, although the differences in telomere length and number were not significant at 2 months of age, suggesting that loss of Mst1/2 promotes telomeric loss and causes premature ageing. In addition, we found that the number of telomere foci in peritoneal macrophages of DKO mice received NAC supplementation in drinking water for 7 months from 1-month old, was significantly increased (Fig. 2i). Taken together, these data indicate that Mst1/2 might play an essential antiageing role in macrophages.

**Mst1/2 sense phagosomal and mitochondrial ROS.** Phagocytes primarily rely on the two main ROS sources, mitochondrial and phagosomal membrane-associated NADPH oxidase machinery, for pathogen killing[8]. Interestingly, in WT macrophages, we observed that Mst1/2 were activated and recruited from the cytosol to phagosomes containing engulfed bacteria and the mitochondrial compartments with super-resolution immunofluorescence microscopy (SIM) approach and immunoblot, respectively (Fig. 3a–d), while treatment with NAC abolished the Mst1/2-phagosome interaction and attenuated Mst1/2 activation (Fig. 3b–d). Similarly, Mst1/2 were recruited from the cytosol to

**Fig. 3** Mst1/2 sense phagosomal and mitochondrial ROS. **a** Flow cytometry analysis of cellular ROS levels in WT BMDMs pretreated with or without NAC, infected with *E. coli* (MOI: 100) and stained with CellRox for 30 min. **b** SIM of Mst1 staining (red) and DAPI-stained nuclei (blue) in WT BMDMs infected with GFP-*E. coli* (green) treated with or without NAC as indicated; ×25 magnification of areas outlined in the main images are shown next to the main images. Scale bars, 20 μm. **c** Immunoblot analysis of phosphorylated (p)-Mob1, Mob1, p-Mst1/2, Mst1, Mst2, and GAPDH in BMDMs pretreated with PBS or NAC (5 μM) and then infected with *E. coli* (MOI: 100). **d** Immunoblot analysis of Mst1, Mst2, β-actin and Hsp60 in the cytoplasmic (Cyto) and mitochondrial (Mito) fractions of NAC-treated or non-treated BMDMs infected with *E. coli* (MOI: 100) for the indicated time. **e** SIM of Mst1 staining (red), Tomm20 (green) and DAPI-stained nuclei (blue) in WT BMDMs treated with DMSO or antimycin A, with or without NAC pretreatment, as indicated; ×49 magnification of areas outlined in the main images are shown next to the main images. Scale bars, 20 μm. **f, g** Immunoblot analysis of Mst1, Mst2, β-actin, and Hsp60 in the cytoplasmic (Cyto) and mitochondrial (Mito) fractions of WT BMDMs treated with antimycin A (**f**) or rotenone (**g**), with or without NAC pretreatment, for the indicated time. **h, i** Immunoblot analysis of p-Mob1, Mob1, p-Mst1/2, Mst1, Mst2, and GAPDH in BMDMs treated with antimycin A (**h**) or rotenone (**i**) for the indicated time or with antimycin A (**h**) or rotenone (**i**) at the indicated dose for 30 min. **j, k** Immunoblot analysis of p-Mob1, Mob1, p-Mst1/2, Mst1, Mst2, and GAPDH in PBS or NAC pretreated BMDMs followed with 10 μM antimycin A (**j**) or 3 μM rotenone (**k**) treatment. **l** Flow cytometry analysis of mitochondrial ROS levels in THP1 cells treated with 10 μM antimycin A or 3 μM rotenone followed with or without Mst1/2 kinases inhibitor, XMU-MP-1, treatment for indicated times. Data are from one experiment representative of three independent experiments with similar results

the mitochondrial membrane when BMDMs were treated with antimycin A or rotenone, which mimics the mitochondrial oxidative burst (Fig. 3e–g). Importantly, NAC treatment effectively disrupted the mitochondrion-Mst1/2 interaction induced by antimycin A or rotenone (Fig. 3e–g). In addition, antimycin A or rotenone treatment profoundly enhanced the phosphorylation levels of Mst1/2 and its downstream substrate Mob1 in a dose- or time-dependent manner (Fig. 3h, i), while NAC treatment blocked this effect (Fig. 3j, k). Furthermore, we found that the Mst1/2 kinase inhibitor, XMU-MP-1, treatment blocked the attenuation of ROS levels in THP-1 macrophages at the late phase of antimycin A treatment (Fig. 3l). These data indicate that Mst1/2 can sense the oxidative burst from mitochondrial or phagosomal NADPH oxidase machinery and that released ROS subsequently activates Mst1/2 in macrophages to protect cells against oxidative stress during anti-infection defense processes.

**Nrf2 is required for the antioxidant response in macrophages.** The induction of antioxidant enzymes is critical for protecting cells against toxic free radicals. We observed that the mRNA and protein levels of various antioxidant genes such as, *Nqo1*, *Ho-1*, *Gclc*, and *Gclm* were dramatically lower in DKO BMDMs than that of WT BMDMs when cells were treated with LPS, antimycin A, rotenone or $H_2O_2$, or infected with *E. coli*, indicating that the enzymatic antioxidant system is defective in Mst1/2-deficient macrophages (Fig. 4a–d). Oxidative stress is involved in the activation of several major transcription factors, such as those in the Nrf2 and FoxO1/3 families[39,40], which activate the expression of the antioxidant genes to promote cellular adaptation to oxidative stress. Interestingly, we found that FoxO1/3 and Nrf2 were differentially expressed in various immune cell types. FoxO1 and FoxO3 were highly expressed in T cells and B cells but were barely detectable in macrophages, whereas Nrf2 was preferentially expressed in macrophages (Fig. 4e, f). We performed immunofluorescence co-staining of FoxO1 or Nrf2 with CD3 or F4/80 in WT mouse spleen sections. The results showed that FoxO1 was exclusively expressed in CD3$^+$ T cells, not in F4/80$^+$ macrophages; in contrast, all F4/80$^+$ cells highly expressed Nrf2, which was detected at a lower level in CD3$^+$ T cells (Fig. 4g). Consistently, basal ROS levels were not affected in FoxO1- or FoxO3-deficient BMDMs but were significantly increased in Nrf2-deficient macrophages compared with their WT counterparts (Fig. 4h, i). Furthermore, Nrf2-knockout ($Nrf2^{-/-}$) macrophages isolated from 8-month-old $Nrf2^{-/-}$ mice exhibited premature ageing, as shown by shorter telomeres and fewer telomere foci than in $Nrf2^{+/+}$ WT cells (Fig. 4j, k). These results suggest that Nrf2 is the key transcriptional activator of the antioxidant response in macrophages.

**Mst1/2 abrogate Keap1-mediated Nrf2 degradation.** Due to the significant downregulation of the expression of genes encoding antioxidant enzymes in DKO macrophages and the indispensable role of the antioxidant transcription factor Nrf2 in macrophages, we next sought to determine the function of Nrf2 in WT or DKO macrophages. We found that both mRNA and protein levels of Nrf2 were increased in WT BMDMs upon the treatment of antimycin A or rotenone in a time-dependent manner (Fig. 5a, b and Supplemental Fig. 2a, b). Interestingly, the accumulation of Nrf2 protein was impaired, although the increased Nrf2 mRNA level was not affected in Mst1/2-deficient BMDMs upon the above-mentioned treatment (Fig. 5a, b and Supplemental Fig. 2a, b), suggesting that Mst1/2 might regulate the protein stability of Nrf2. Previous studies showed that Nrf2 is predominantly degraded through the ubiquitination-mediated proteasome pathway. Under unstressed conditions, Keap1, a substrate adapter subunit of a Cullin 3 (Cul3)-based ubiquitin E3 ligase, interacts with Nrf2 and promotes the ubiquitination and proteasomal degradation of Nrf2[41–43]. Indeed, we observed that Nrf2 ubiquitination was increased in Mst1/2-deficient BMDMs, and antimycin A treatment decreased the Nrf2 ubiquitination level in WT BMDMs, but not in DKO BMDMs (Fig. 5c). Notably, we immunoprecipitated Flag-tagged Mst1 or Mst2 from lysates of THP1 macrophages and performed mass spectrometry, which identified several novel Mst1/2-interacting proteins, including Keap1 (Supplementary Fig. 2c). Interestingly, overexpression of WT Mst2, but not kinase-dead Mst2 (Mst2K/R), attenuated the effect of Keap1 on Nrf2 ubiquitination (Fig. 5d). Moreover, we found that Keap1 can bind to Mst2 and their interaction was enhanced upon *E. coli* infection or antimycin A treatment (Fig. 5e and Supplementary Fig. 2d). We also observed that Keap1 was recruited with Mst1/2 from the cytosol to the mitochondrial membrane when WT BMDMs were treated with *E. coli* or antimycin A (Fig. 5f and Supplementary Fig. 2e). Consistently, with SIM approach, Mst1 protein was condensed and formed large aggregates after *E. coli* infection (from 5 min and at least last for 60 min), and Keap1 co-localized with Mst1 after *E. coli* infection at 5/15/30 min time points, but not at 60 min in BMDM (Fig. 5g and Supplementary Fig. 2f). In contrast, these phenomena were not observed in NAC-treated BMDMs (Fig. 5g).

In determining how Mst1/2 regulates the function of Keap1, we found that Mst2 directly binds to and phosphorylates Keap1 at the amino terminus (amino acids 1–310), as visualized by the "up-shifted" bands using the Phos-tag labeling system followed by sodium dodecyl sulfate polyacrylamide gel electrophoresis (SDS-PAGE) (Fig. 6a, b). Mass spectrometry and site-directed mutagenesis further revealed that Keap1 was phosphorylated by

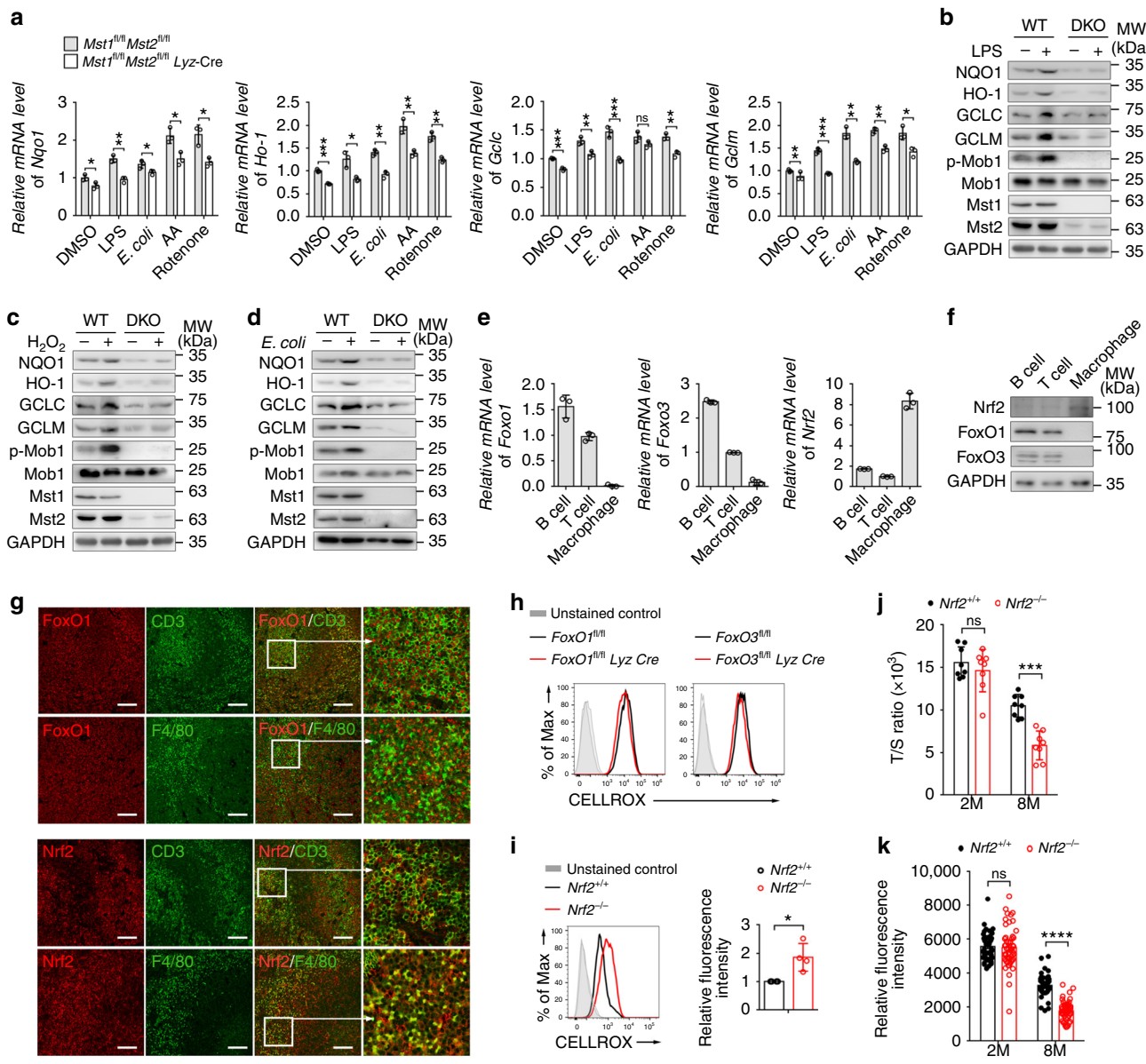

**Fig. 4** Dysregulation of Nrf2 in Mst1/2-null macrophages. **a** RT-qPCR analysis of the mRNA levels of the indicated antioxidant genes in $Mst1^{fl/fl}Mst2^{fl/fl}$ (WT) and $Mst1^{fl/fl}Mst2^{fl/fl}$ Lyz-Cre (DKO) BMDMs infected with *E. coli* (MOI: 100) or treated with LPS, antimycin A or rotenone. **b–d** Immunoblot analysis of GCLC, GCLM, NQO1, HO-1, p-Mob1, Mob1, Mst1, Mst2, and GAPDH in WT and DKO BMDMs treated with LPS (**b**), $H_2O_2$ (**c**) or infected with *E. coli* (MOI: 100) (**d**). **e, f** RT-qPCR analysis (**e**) and immunoblot analysis (**f**) of the *Foxo1*, *Foxo3*, and *Nrf2* gene expression in WT B cells, T cells and macrophages. **g** Confocal microscopy of the distribution of FoxO1, Nrf2, CD3, and F4/80 immunostaining in the indicated colors in spleen sections from WT mice. ×16 magnification of areas outlined in the main images are shown next to the main images. Scale bars, 200 μm. **h, i** Flow cytometry analysis of ROS levels of FoxO1−, FoxO3− (**h**) and Nrf2-deficient (**i**) BMDMs and their WT counterparts. The fluorescence intensity of the ROS dye in the indicated cell samples is shown in the right panel (**i**). **j, k** The relative telomere length (T/S ratio) (**j**) and relative fluorescence intensity of telomere FISH (**k**) in peritoneal macrophages isolated from 2- or 8-month-old $Nrf2^{+/+}$ and $Nrf2^{-/-}$ mice. ns, not significant ($P > 0.05$), *$P < 0.05$, **$P < 0.01$, ***$P < 0.001$, ****$P < 0.0001$ compared with control (Student's *t* test). Data are from one experiment representative of three independent experiments with similar results (mean and s.d. of $n = 3$ (**a, e**, experimental replicates); $n = 4$ (**i**) or $n = 7$ (**j**), biological replicates; mean and s.e.m. of $n = 50$ (**k**), experimental replicates)

Mst1/2 at four amino acid residues (i.e., T51, S53, T55, and T80), which are next to or within the BTB domain (amino acids 77–149) (Fig. 6c, d; Supplementary Fig. 3). To further characterize the effects of Mst1/2-mediated Keap1 phosphorylation on Nrf2 stability, we generated constitutive phospho-mimic or nonphosphorylatable Keap1 mutants by replacing the above-mentioned four T/S residues with aspartate (named Keap1$^{4D}$) or alanine (named Keap1$^{4A}$), respectively. Coexpression of Nrf2 with WT Keap1 (Keap1$^{WT}$), Keap1$^{4A}$ or Keap1$^{4D}$ in the presence of Cul3 revealed that Nrf2 ubiquitination levels were enhanced by Keap1$^{4A}$ but attenuated by Keap1$^{4D}$ (Fig. 6e). We then determined the effect of WT or mutant Keap1 on Nrf2 protein stability by cycloheximide (CHX) assays. Nrf2 levels were dramatically decreased after 3 h of CHX treatment in the presence of Keap1$^{WT}$; Nrf2 degradation was much faster in cells transfected with Keap1$^{WT}$ than in those transfected with empty vector control (Fig. 6f, g). The enhancement of Nrf2 degradation was more pronounced in cells transfected with Keap1$^{4A}$ (Fig. 6f). In contrast, Nrf2 levels in Keap1$^{4D}$-transfected cells were still detectable until 12 h of CHX treatment (Fig. 6g). These results

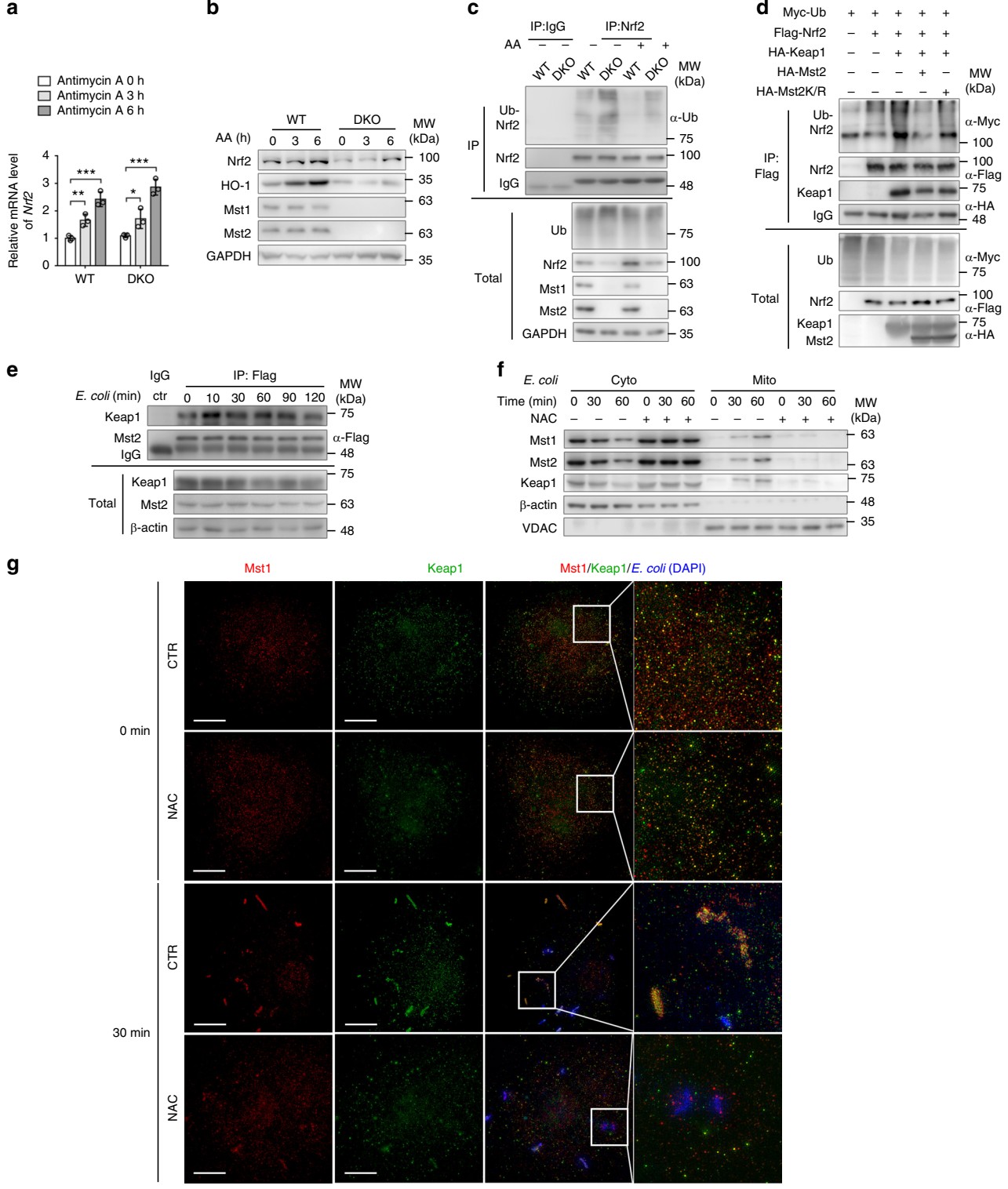

**Fig. 5** Mst1/2 downregulates Keap1-mediated ubiquitination of Nrf2. **a**, **b** RT-qPCR analysis of the mRNA levels of *Nrf2* (**a**) and immunoblot analysis of Nrf2, HO-1, Mst1, Mst2, and GAPDH (**b**) in *Mst1*[fl/fl]*Mst2*[fl/fl] (WT) and *Mst1*[fl/fl]*Mst2*[fl/fl] *Lyz*-Cre (DKO) BMDMs treated with 10 μM antimycin A (AA) for indicated times. *$P < 0.05$, **$P < 0.01$, ***$P < 0.001$, compared with the control sample (Student's *t* test, $n = 3$, experimental replicates). **c** Immunoblot analysis of Nrf2 ubiquitination in total lysates (bottom) and anti-IgG or anti-Nrf2 immunoprecipitates (IP, top) of WT and DKO BMDMs treated with or without antimycin A, probed with anti-ubiquitin (α-Ub) and antibodies to Nrf2, Mst1, Mst2, and GAPDH. **d** Immunoassay of the ubiquitination of Nrf2 in anti-Flag IP (top) and total lysates (bottom) of 293T cells expressing various combinations (above lanes) of Myc-tagged Ub, Flag-tagged Nrf2, HA-tagged Keap1, Mst2 or Mst2K/R, and Myc-tagged Ub. **e** Immunoblot analysis of Keap1 and Mst2 in anti-IgG or anti-Flag IP (top) and total lysates (bottom) of WT BMDM cell line stably expressing Flag-tagged Mst2 infected with *E. coli* (MOI: 100) for indicated times. **f** Immunoblot analysis of Mst1, Mst2, β-actin, and VDAC in the cytoplasmic (Cyto) and mitochondrial (Mito) fractions of WT BMDMs pretreated with or without NAC and infected with *E. coli* (MOI: 100) for indicated times. **g** SIM of the co-localization of Mst1 (red) and Keap1 (green) in WT BMDMs treated with or without NAC and infected with *E. coli* (blue) for 30 min; ×16 magnification of areas outlined in the main images are shown next to the main images. Scale bars, 20 μm. Data are from one experiment representative of three independent experiments with similar results

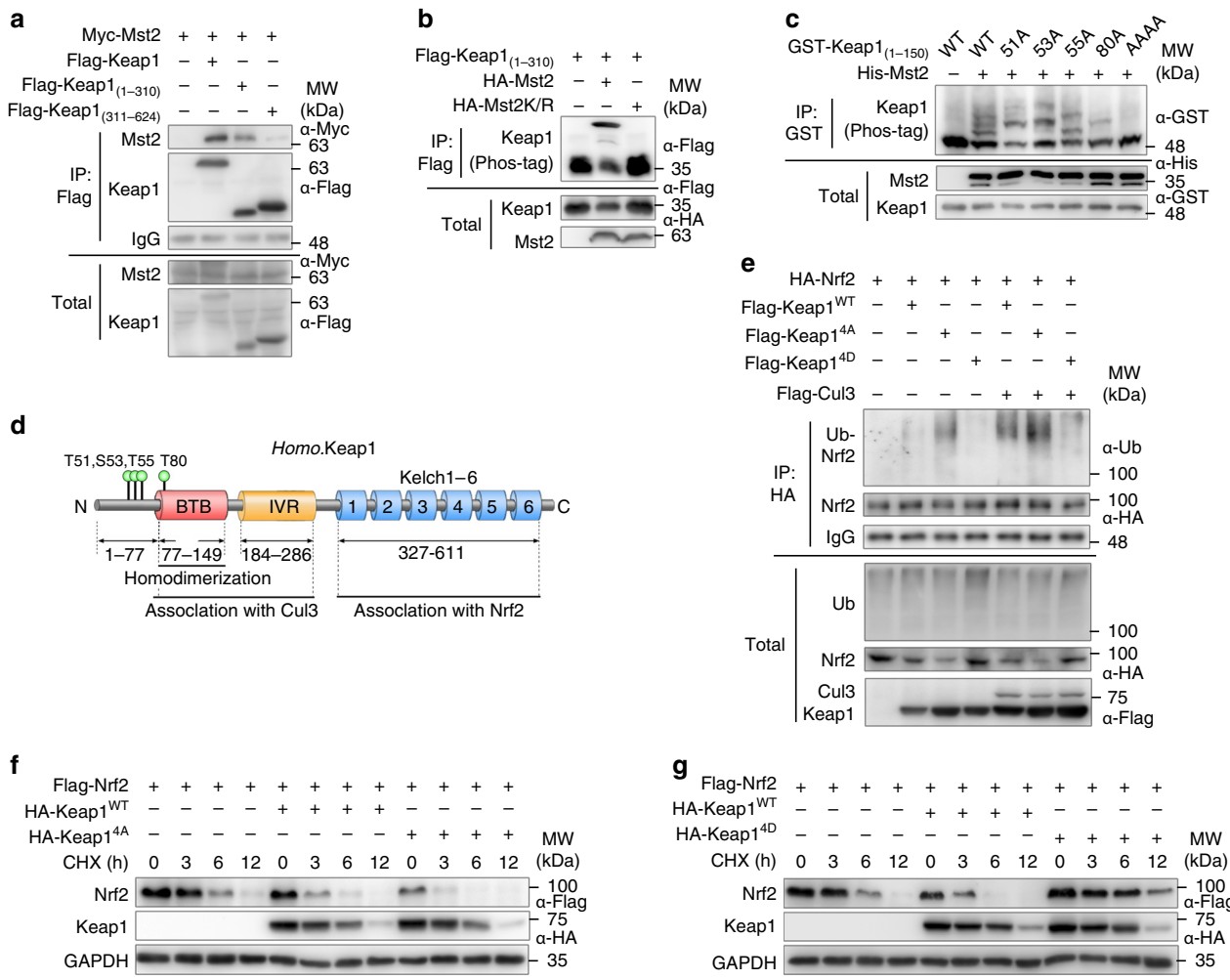

**Fig. 6** indentification of Mst1/2-mediated phosphorylation sites on Keap1. **a** Immunoblot analysis of Mst2, Keap1 in total lysates (bottom) and anti-Flag IP (top) of 293T cells expressing various combinations (above lanes) of Flag-tagged full length Keap1, truncated Keap1$_{(1-310)}$ or Keap1$_{(311-624)}$ and Myc-tagged Mst2. **b** Phos-tag and SDS-PAGE analysis of Flag-tagged Keap1 (1–310) expressed together with HA-tagged wild-type Mst2 or kinase-inactive Mst2K/R in 293T cells immunoprecipitated with anti-Flag conjugated beads. **c** Phos-tag and SDS-PAGE analysis of GST-tagged wild-type or various mutated Keap1$_{(1-150)}$ expressed together with His-tagged Mst2 in 293T cells. **d** Mst1/2 kinases-mediated phosphorylation sites on the Keap1 showing the domains of BTB, IVR, and Kelch1-6; numbers below diagrams indicate amino acid range of each domain. **e** Immunoassay of the ubiquitination of Nrf2 in anti-HA IP (top) and total lysates (bottom) of 293T cells expressing various combinations (upper lanes) of Flag-tagged Cul3, Flag-tagged WT, mutated Keap1$^{4A}$ or Keap1$^{4D}$, and HA-tagged Nrf2. **f, g** Immunoblot analysis of total lysates in 293T cells expressing various combinations (above lanes) of HA-tagged wild-type Keap1$^{WT}$, mutated Keap1$^{4A}$ (**f**) or Keap1$^{4D}$ (**g**) and Flag-tagged Nrf2 followed by CHX treatments with the indicated time. Data are from one experiment representative of three independent experiments with similar results

suggest that Mst1/2 prevent Nrf2 ubiquitination and degradation by phosphorylating Keap1.

**Phosphorylated Keap1 fails to form dimer or polymer.** The Mst1/2-mediated phosphorylation sites on Keap1 are located within or next to the BTB domain, which are essential for binding the E3 ubiquitin ligase Cul3 or Keap1 dimerization or polymerization (Fig. 6d). We found that Mst1/2-mediated Keap1 phosphorylation did not affect the interaction between Keap1 and Cul3, as shown by the equal binding ability of Cul3 to Keap1$^{WT}$, Keap1$^{4A}$, and Keap1$^{4D}$ in 293T cells co-expressing Cul3 with Keap1$^{WT}$, Keap1$^{4A}$, or Keap1$^{4D}$ (Fig. 7a). We then performed co-immunoprecipitation assays using 293T cells cotransfected with HA- and Flag-tagged Keap1$^{WT}$ or mutant Keap1 (Keap1$^{4A}$ or Keap1$^{4D}$) to determine whether Mst1/2-mediated Keap1 phosphorylation affects Keap1 polymerization. The results showed that Keap1$^{WT}$, as well as Keap1$^{4A}$, were able to form protein

dimers/polymers, whereas the constitutive phospho-mimic Keap1$^{4D}$ was unable to polymerize (Fig. 7b). In addition, native PAGE analysis confirmed that the phospho-mimic Keap1$^{4D}$ or Keap1$^{WT}$ cotransfected with Mst2 formed less dimer/polymer (Fig. 7c, d). The polymerization of WT and various mutant forms of Keap1 was further visualized with SIM approach (Fig. 7e, f). Notably, HA-tagged Keap1$^{4A}$ (green) and Myc-tagged Keap1$^{4A}$ (red) were able to form large foci (Fig. 7e, the top panels). In contrast, only some Myc-tagged Keap1$^{WT}$ and much less Keap1$^{4D}$ co-localized with HA-tagged Keap1$^{4A}$ to form large aggregates (Fig. 7e, the lower two panels). These observations indicate that Mst1/2-mediated Keap1 phosphorylation disrupts the dimerization or polymerization of Keap1. Interestingly, the immunoprecipitation assay showed that, compared with Keap1$^{WT}$ or Keap1$^{4D}$, Keap1$^{4A}$ had a higher binding affinity for Nrf2 (Fig. 7g), suggesting that dimerization or polymerization of keap1 might cause a conformation change for better binding Nrf2. Consistently, the enhanced association of endogenous

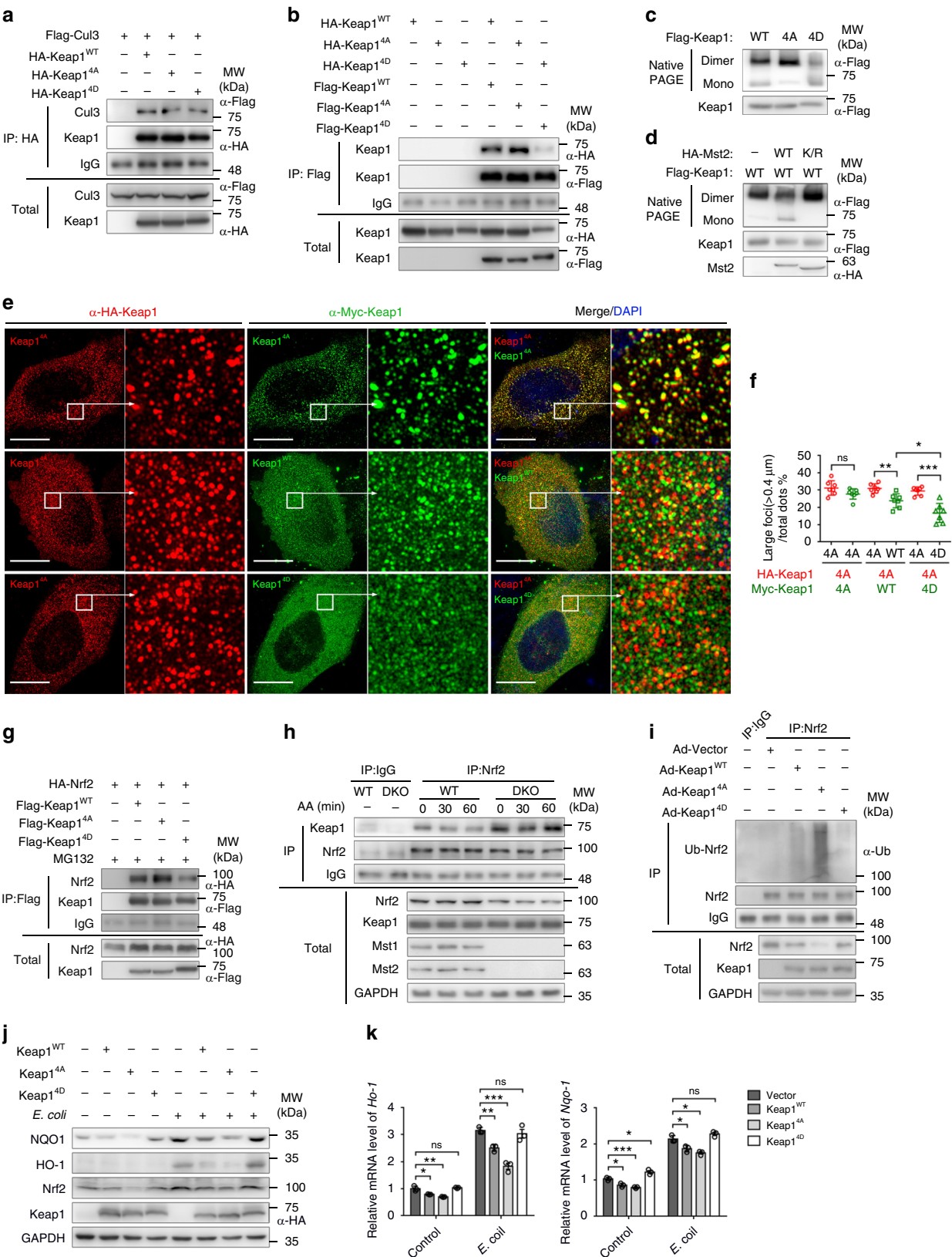

Keap1 and Nrf2 was also found in Mst1/2-deficient BMDMs and antimycin A treatment attenuated the association of Keap1 and Nrf2 in WT BMDMs, but not in DKO BMDMs (Fig. 7h). These results indicated that Mst1/2-mediated Keap1 phosphorylation disrupts the dimerization or polymerization of Keap1 and the interaction between Keap1 and Nrf2.

To verify the effects of Mst1/2-mediated Keap1 phosphorylation in macrophages, WT BMDMs were infected with adenovirus expressing Keap1$^{WT}$, Keap1$^{4A}$, or Keap1$^{4D}$ and determined the ubiquitination level of Nrf2. Indeed, overexpression of Keap1$^{4A}$, which mimics no Mst1/2-mediated phosphorylation and forms abundant dimers/polymers, dramatically enhanced the

**Fig. 7** Mst1/2-mediated Keap1 phosphorylation disrupts the Keap1-Nrf2 complex. **a** Immunoblot analysis of Cul3 and Keap1 in total lysates (bottom) and anti-HA IP (top) of 293T cells expressing various combinations of Flag-tagged Cul3, HA-tagged Keap1$^{WT}$, Keap1$^{4A}$ or Keap1$^{4D}$. **b** Immunoblot analysis of keap1 levels in total lysates (bottom) and anti-Flag IP (top) of 293T cells expressing various combinations of HA-tagged or Flag-tagged wild-type Keap1$^{WT}$, mutated Keap1$^{4A}$ or Keap1$^{4D}$. **c, d** Immunoblot analysis of dimerization of Keap1 in total lysates of 293T cells expressing Flag-tagged WT or mutated Keap1 (**c**) or various combinations of HA-tagged WT or kinase inactivated Mst2K/R with Flag-tagged Keap1 (**d**) using native or regular PAGE. **e** SIM of HeLa cells cotransfected to express HA-tagged Keap1$^{4A}$ (red) with Myc-tagged Keap1$^{4A}$, Keap1$^{WT}$ or Keap1$^{4D}$ (green); ×64 magnification of areas outlined in the main images are shown next to the main images. Scale bars, 20 μm. **f** Quantification of the percentage of foci (larger than 0.4 μM) on e.n.s. no significant, *$P < 0.05$, **$P < 0.01$, ***$P < 0.001$ (Student's t test, mean and s.d. of $n = 7$ images). **g** Immunoblot analysis Nrf2 and Keap1 in total lysates (bottom) and anti-Flag IP (top) of 293T cells expressing various combinations of Flag-tagged Nrf2 and HA-tagged Keap1$^{WT}$, Keap1$^{4A}$, or Keap1$^{4D}$ in present of MG132. **h** Immunoblot analysis of Keap1 and Nrf2 in anti-IgG or anti-Nrf2 IP (top) and total lysates (bottom) of Mst1$^{fl/fl}$Mst2$^{fl/fl}$ (WT) and Mst1$^{fl/fl}$Mst2$^{fl/fl}$ Lyz-Cre (DKO) BMDMs treated with antimycin A (AA) for indicated times, probed with antibodies to Keap1, Nrf2, Mst1, Mst2, and GAPDH. **i** Immunoassay of the ubiquitination of Nrf2 in anti-IgG or anti-Nrf2 IP (top) and total lysates (bottom) of BMDMs infected with adenovirus expressing (Ad−) Keap1$^{WT}$, Keap1$^{4A}$, Keap1$^{4D}$ or control vector. **j, k** Immunoblot analysis of NQO1, HO-1, Nrf2, and Keap1 (**j**) or RT-qPCR analysis of Nqo1 and Ho-1 (**k**) in Raw264.7 cells expressing Flag-tagged Keap1$^{WT}$, Keap1$^{4A}$, or Keap1$^{4D}$ infected with E. coli (MOI: 100). ns, not significant ($P > 0.05$); *$P < 0.05$, **$P < 0.01$, ***$P < 0.001$, compared with the control sample (Student's t test) (mean and s.d. of $n = 3$, experimental replicates). Data are from one experiment representative of three independent experiments with similar results

---

ubiquitination level of Nrf2 and decreased Nrf2 protein levels, while overexpression of nonpolymerizable Keap1$^{4D}$ led to the lower Nrf2 ubiquitination level and no or little effect on the Nrf2 protein levels (Fig. 7i). Consistently, overexpressing Keap1$^{4A}$, but not Keap1$^{WT}$ or Keap1$^{4D}$ also profoundly downregulated the Nrf2 protein levels, as well as the mRNA and protein levels of Nqo1 or Ho-1 genes, in Raw264.7 cells with E. coli infection (Fig. 7j, k). Therefore, our data suggested that Mst1/2 can stabilize the key antioxidant factor Nrf2 by phosphorylating Keap1 and blocking its polymerization and thus its ability to target Nrf2 for degradation in macrophages.

**Overexpression of Nrf2 rescues the Mst1/2-null phenotype.** We have shown that kinases Mst1/2 are required for maintaining Nrf2 protein stability. To further determine the role of Mst1/2-mediated Nrf2 protein stability in Mst1/2-mediated antioxidant defense in macrophages, DKO BMDMs were infected with control adenovirus expressing GFP (Ad-GFP) or adenovirus expressing Nrf2 (Ad-Nrf2). Indeed, reintroduction of Ad-Nrf2, but not of Ad-GFP, significantly downregulated basal ROS in DKO BMDMs (Fig. 8a). Consistently, Ad-Nrf2, but not control Ad-GFP, significantly increased the transcription of the Ho-1 and Nqo1 genes and decreased p-H2A.X levels in DKO BMDMs with or without H$_2$O$_2$ treatment (Fig. 8b, c). In addition, Ad-Nrf2, but not Ad-GFP significantly decreased the percentage of apoptotic events in DKO BMDMs (Fig. 8d, e). To further characterize the function of Nrf2 in Mst1/2 deficiency-induced premature ageing, we performed transplantation experiments by injecting Mst1/2-deficient bone marrow cells transduced with retrovirus expressing Nrf2 or control GFP into lethally irradiated Mst1$^{fl/fl}$Mst2$^{fl/fl}$ mice. Six months later, GFP-positive peritoneal macrophages were sorted and subjected to telomere foci analysis by FISH. Reintroduction of Nrf2 prevented telomeric loss, as shown by the significantly increased relative fluorescence intensity of telomere foci in Mst1/2-deficient macrophages (Fig. 8f). Taken together, these results suggest that the Mst1/2-Nrf2 axis is essential for maintaining redox balance and preventing premature ageing in macrophages (Fig. 8g).

**Discussion**

Large amounts of ROS are produced by host phagocytes and exert antimicrobial activity against a broad range of pathogens[6,9]. However, how phagocytes sense intracellular ROS levels and achieve self-protection against oxidative stress-induced damage and ageing remain incompletely understood. Here, we identified a previously unknown aspect of antioxidant and anti-ageing

signaling involved in host defense: the role of Mst1/2 in sensing ROS and enhancing the stability and activity of Nrf2, the key antioxidant transcription factor for ROS scavenging and anti-ageing processes. We found that either phagosomal or mitochondrial ROS release attracted kinases Mst1/2 from the cytosol to the phagosomal or mitochondrial membrane and subsequently activated Mst1/2, which in turn regulates Nrf2 protein stability. Notably, ROS triggered the activation of Mst1/2 to phosphorylate the substrate adapter protein Keap1 in the Keap1-Cul3-Rbx1 E3 ubiquitin ligase complex to prevent its polymerization and association with Nrf2, thereby blocking the targeting of Nrf2 for degradation. Deletion of Mst1/2 greatly impaired the ability of macrophages to resist oxidative stress and the ageing process. Importantly, the reintroduction of Nrf2 abrogated Mst1/2 deficiency-induced phagocyte ageing and death. Thus, our results identify the Mst-Nrf2 axis as an important ROS sensing, antioxidant, and anti-ageing mechanism in phagocytes during an antimicrobial response.

Hydrogen peroxide is commonly used to activate Mst1/2 in vitro and several mechanisms of ROS-mediated Mst1/2 activation have been proposed[14,19,20,34,44–48]. However, it is not clear whether model systems exposed to exogenous hydrogen peroxide have relevance to systems in which the oxidant is generated endogenously. Here, we demonstrated that oxidants generated endogenously by phagosomes or mitochondria recruited Mst1/2 to phagosomes or mitochondria and markedly activated these kinases. The antioxidant NAC disrupted the association of Mst1/2 with phagosomes or mitochondria. We further revealed that oxidant sensing and signaling by Mst1/2 were important for attenuating oxidative stress. Moreover, we recently found that Mst1/2 are required to promote the mitochondria-phagosome interaction to produce the ample amounts of phagosomal and mitochondrial ROS necessary for the bactericidal activity of phagocytes[33]. Collectively, Mst1/2 function as critical coordinators of ROS generation and scavenging in phagocytes during an antimicrobial response. It will be interesting to identify the mechanism underlying Mst1/2-mediated oxidant sensing and to determine whether endogenous hydrogen peroxide modulates Mst1/2 signaling by altering the oxidation status of kinases Mst1/2.

Oxidative stress is involved in the activation of two major transcription factors, Nrf2 and FoxO1/3, but the exact underlying molecular mechanisms remain elusive[42,49,50]. In the nervous system, oxidative stress induces the Mst1-mediated phosphorylation of FoxO3 at Ser207, leading to the release of FoxO3 from 14-3-3 proteins and the consequent accumulation of FoxO3 in the nucleus, where FoxO3 induces the expression of antioxidant

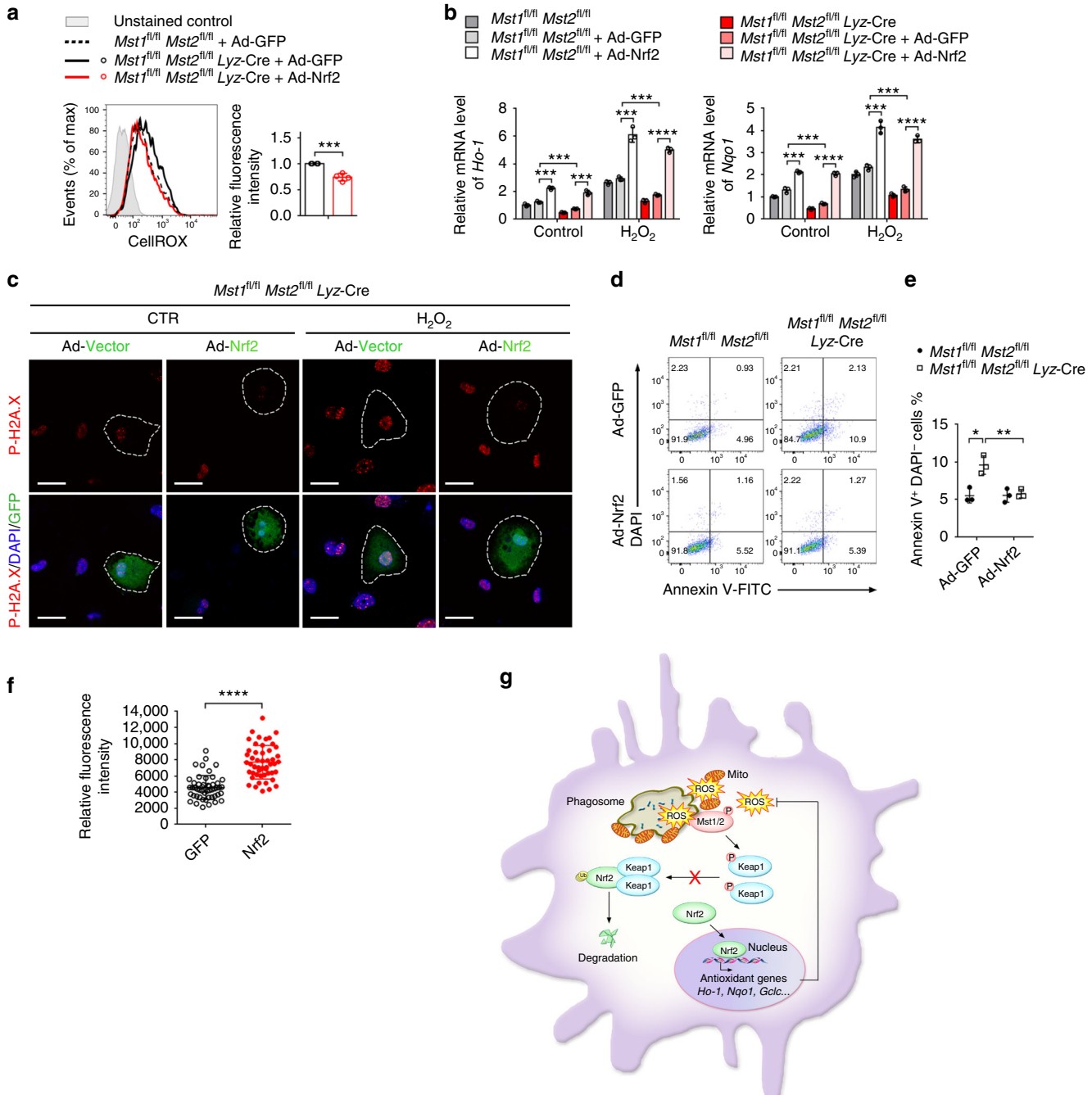

**Fig. 8** Reintroduction of Nrf2 rescues the Mst1/2-null phenotype. **a** Flow cytometry analyzing of ROS levels in $Mst1^{fl/fl}Mst2^{fl/fl}$ (WT) BMDMs, $Mst1^{fl/fl}Mst2^{fl/fl}$ $Lyz$-Cre (DKO) BMDMs, or DKO BMDMs infected with adenovirus expressing GFP (Ad-GFP) or Nrf2 (Ad-Nrf2), with the CellRox dye and quantification of the relative fluorescence intensity in the indicated cell samples shown in the right panel. **b–e** RT-qPCR analysis of the mRNA levels of the antioxidant genes $Ho$-1 and $Nqo$-1 (**b**), fluorescence microscopy of p-H2A.X (red), DAPI-stained nuclei (blue) and GFP+ cells (green) (**c**), Annexin V/DAPI staining (**d**) and quantification of Annexin V+DAPI− cells (**e**) in WT BMDMs, DKO BMDMs, or DKO BMDMs infected with Ad-GFP or Ad-Nrf2, followed with or without $H_2O_2$ treatment as indicated. Scale bars, 20 μm. **f** Relative fluorescence intensities of telomere FISH of GFP+ peritoneal macrophages isolated from WT mice transplanted for 6 months with DKO bone marrow cells, which were transduced with retrovirus expressing Nrf2, or control GFP, respectively. **g** A proposed working model for kinases Mst1/2 sense ROS and maintain cellular redox balance by modulating the stability of Nrf2. Phagosomal or mitochondrial ROS release attracts Mst1/2 to cap around phagosome or mitochondrion from the cytosol and activates Mst1/2; Mst1/2 phosphorylate Keap1 to stabilize Nrf2 and regulate the expression of antioxidant enzymes to protect cell against oxidative damage. Data are from one experiment representative of three independent experiments with similar results. ns, not significant ($P > 0.05$); *$P < 0.05$, **$P < 0.01$, ***$P < 0.001$, ****$P < 0.0001$, compared with the control sample (Student's $t$ test). Data are from one experiment representative of three independent experiments with similar results (mean and s.d. of $n = 3$ (**a**, **b**, **e**); mean and s.e.m. of $n = 50$ (**f**); biological replicates (**a**, **e**); and experimental replicates (**b**, **f**))

or apoptotic genes and thereby leads to either cell recovery or cell death in response to oxidative stress[10,11,46,47]. Although the antioxidant role of the Mst1-FoxO signal in the nervous system has been extensively studied, the function of Mst1-mediated cellular responses to oxidative stress in the immune system need to be further defined. Some studies showed that the FoxO1/3 protein levels were dramatically decreased in *Mst1*-deficient T cells while others demonstrated that no significant change in the FoxO1/3 protein levels were observed in the naive T cells and Tregs of *Mst1*$^{-/-}$ mice[24,29,51,52]. Thus, the role of FoxO1/3 in Mst1-mediated oxidative responses and cell death in T lymphocytes remained defined. Interestingly, we found that FoxO1/3 and Nrf2 are differentially expressed in various immune cell types. FoxO1/3 is highly expressed in T cells and B cells but barely detectable in macrophages, whereas Nrf2 is preferentially expressed in macrophages. We further demonstrated that Mst1/2 might function as molecular switches to activate the antioxidant transcription factor Nrf2 and suppress excess ROS generation to attenuate ROS-induced damage and the ageing process in macrophages. In line with these findings, we conclude that the Mst-FoxO/Nrf2 signaling cascade plays an important role in cellular responses to oxidative stress and might contribute to pathological states, including infection and neurodegenerative diseases in the elderly.

Age-related innate immune changes put older adults at a greater risk of impairment and death from infection, such as influenza or pneumonia[53]. Elucidation of Mst-Nrf2 signaling as an oxidative stress response mechanism in macrophages suggests that this pathway may provide a mechanistic basis for how oxidative stress contributes to the pathogenesis of ageing-associated inflammation and infection. Aged macrophages have an impaired respiratory burst as a result of altered intracellular signaling, rendering them less able to destroy bacteria. Gaining an understanding of the functional integrity of the innate immune system in aged subjects is of paramount importance. The mechanisms that underlie this heightened ageing-associated basal inflammation also seem to involve other changes in innate immune cells, including altered expression of pattern recognition receptors (PRRs), activation of PRRs by endogenous ligands associated with cellular damage, and aberrant signaling events downstream of PRR activation that lead to cytokine secretion[1]. Because of the critical roles of Mst1/2 in innate immunity[32,33,54–62], gaining a better understanding of the functions and molecular mechanisms of Mst1/2-mediated signaling in the altered activation of PRRs during the ageing process is of great interest in the fields of immunology and ageing.

## Methods

**Animals**. The conditional knockout (KO) of Mst1 and Mst2 has been described previously[63]. WT C57BL/6, *Lyz2*-Cre mice (004781) and B6.129 × 1-*Nfe2l2*$^{tm1Ywk}$/J (*Nrf2*$^{-/-}$, 017009), *Foxo1*$^{tm1Rdp}$/J (*Foxo1*$^{fl/fl}$, 024756), *Foxo3*$^{tm1Rdp}$/J (*Foxo3*$^{fl/fl}$, 024668) were originally from the Jackson Laboratory. All mice were maintained under specific pathogen-free conditions at the Xiamen University Laboratory Animal Center. These mouse experiments were approved by the Institutional Animal Care and Use Committee and were in strict accordance with good animal practice as defined by the Xiamen University Laboratory Animal Center.

**Chemicals and reagents**. N-Acetyl-ʟ-cysteine (NAC, A9165), LPS, rotenone, antimycin A, mg132 (C2211), His beads (P6611) and ANTI-FLAG® M2 Affinity Gel (A2220) were from Sigma-Aldrich. The nuclear and cytoplasmic protein extraction kit was from Beyotime Biotech. Blue Native PAGE Sample Buffer (C506055-0005), native PAGE (C61104-0001) and oligo nucleotides were synthesized by Sangon Biotech. The Phos-tag TM Acrylamide AAL-107 was from the NARD Institute.

**Flow cytometry assays**. Single cells isolated from the bone marrow, spleen or peritoneal cavity were stained for 30 min with the appropriate fluorescence-conjugated antibodies and washed, then were resuspended with flow cytometry staining buffer (2% fetal bovine serum (FBS) in phosphate-buffered saline (PBS))

containing DAPI (4′,6-diamidino-2-phenylindole; Invitrogen). Stained cells were analyzed with a BD LSRFortessa flow cytometer (BD Biosciences). Flow cytometry data were plotted and quantified, through the use of median fluorescence intensity, with FlowJo software (TreeStar) (Supplementary Fig. 4). Fluorescence-conjugated anti-CD11b (M1/70), anti-F4/80 (BM8) and Annexin V, and Annexin V binding buffer (422201) were from BioLegend.

**ROS measurement**. Cells (BMDMs) were plated in non–tissue-culture-treated dishes. Samples were treated with NAC (5 μM) for 30 min as needed, followed by the treatment of other indicated stimulants (10 μM antimycin A or 3 μM rotenone). The culture medium was removed and then the cells were washed with PBS and then incubated for 30 min at 37 °C with MitoS OX (for measurement of mROS superoxide; Invitrogen) and/or CM-H2DCFDA or CellROX (for measurement of total cellular H$_2$O$_2$; Invitrogen) at a final concentration of 5 μM in serum-free DMEM (Invitrogen). The cells were washed with warmed PBS, removed from the plates by pipetting with 1% trypsin containing 1 mM EDTA, pelleted at 1600 r.p.m. for 3 min, immediately resuspended in cold PBS containing 1% FBS and analyzed by flow cytometry (Supplementary Fig. 4a).

**Cell culture**. The 293T (CRL-11268), HeLa (CCL-2), RAW264.7 (TIB-71), and THP1 (TIB-202) cell lines from the American Type Culture Collection (ATCC) were tested for mycoplasma contamination and were found to be negative, then were cultured in DMEM supplemented with 10% FBS and 1× penicillin–streptomycin (Invitrogen).

For BMDMs, the femur and tibia were collected from mice of each genotype, and bone marrow cells were flushed with complete DMEM containing 50 μg ml$^{-1}$ streptomycin and 10% FBS. Erythrocytes were removed via treatment with red blood cell lysis buffer (Sigma), and the cell suspensions were filtered through a 40-μm cell strainer for the removal of any cell clumps. The single-cell suspensions were then cultured for 1 h at 37 °C, and nonadherent cells were collected and replated in complete DMEM with 25% medium conditioned by L929 mouse fibroblasts. For full differentiation of BMDMs, the cells were cultured for an additional 8 days with replacement of the medium every 2 days. All cells were CD11b⁺F4/80⁺ when analyzed by flow cytometry.

**Cell mitochondria isolation**. Cell mitochondria were purified with the Cell Mitochondria Isolation Kit according to the manufacturer's instructions (Beyotime Biotech, C3601). Briefly, cold PBS washed cell pellets (5 × 10$^6$) were resuspended in 200 μl of Mitochondria Isolate Reagent, incubated on ice for 10–15 min and then homogenized with a glass homogenizer. The homogenate was centrifuged at 600 g (1000 g for highly purified mitochondria) for 10 min at 4 °C. The supernatant was transferred to a new tube and spun at 11,000g (3500g for highly purified mitochondria) for 10 min to pellet mitochondria. The resulting cytosolic supernatant was saved for western blot analysis. The mitochondrial pellet was washed with cold PBS and resuspended in lysis buffer for western blotting. The purity of fractions was confirmed by using β-actin (cytosolic fraction, cyto.) and Hsp60 or VDAC (mitochondrial fraction, mito.).

**Real-time RT-PCR**. Following isolation with TRIzol reagent (Invitrogen), mRNA was specifically purified with an RNAeasy Mini Kit (Qiagen). First-strand cDNA was then obtained with the PrimeScript RT reagent Kit with gDNA Eraser (Takara). Real-time quantitative PCR as performed using SYBR Premix Ex TaqII (Takara) and the Bio-Rad iCycler iQ system (Bio-Rad, Hercules, CA, USA). All runs were accompanied by the internal control gene *Gapdh*. The samples were run in triplicate and normalized to *Gapdh* using a ΔΔ cycle threshold-based algorithm, to provide arbitrary units representing relative expression levels. The primer sequences for specific genes are listed in Supplementary Table 1.

**Ubiquitination assay**. 293T cells were transfected with the appropriate plasmids for 36 h and were lysed in ice-cold lysis buffer ("TNTE 0.5%": 50 mM Tris-HCl, pH 7.5, 150 mM NaCl, 1 mM EDTA and 0.5% Triton X-100, containing 10 mM NaF, 2 mM Na3VO4, 10 mg ml$^{-1}$ leupeptin and 1 mM PMSF). The cell lysates were subjected to immunoprecipitation with anti-Flag or anti-HA as indicated, were eluted by boiling 10 min in 1% SDS, were diluted ten times in lysis buffer TNTE 0.5% and then underwent re-immunoprecipitation with anti-Flag (2× IP). The ubiquitin-conjugated proteins were detected by immunoblot analysis with the appropriate antibodies (identified below).

**PAGEs and immunoblotting**. Gels for SDS-PAGE or Phos-tag SDS-PAGE were prepared according to the manufacturer's instructions (NARD Institute). Proteins were separated by SDS-PAGE or Phos-tag SDS-PAGE. For Native-PAGE, the cells were harvested by the Blue Native PAGE sample buffer (Sangon, C506055-0005); and samples were separated by Native-PAGE (Sangon, C601104-0001). Separated proteins were transferred onto a PVDF membrane and then were identified by immunoblot analysis with the appropriate primary antibodies at a dilution of 1:1000 (or as otherwise stated below). Antibody to phosphorylated (p−) H2A.X (9718), anti-H2A.X (3636), anti-Caspase3 (9664), anti-Mst1 (3682), anti-Mst2 (3952), anti-GAPDH (5174), anti-FoxO1 (2880), anti-FoxO3 (12829), anti-α-

tubulin (2144), anti-ubiquitin (3936), anti-Nrf2 (12721, for IP), anti-Rabbit IgG (2729), and anti-Flag (14793) were from Cell Signaling Technology. Anti-PARP (13371-1-AP), anti-Keap1 (10503-2-AP), and anti-HSP60 (66041-1-lg) were from Proteintech. Anti-Nrf2 (ab62352 for WB; ab31163 for IF) were from Abcam. Anti-HA (sc-7392), anti-HA (sc-805), anti-Myc (sc-40), and anti-GST (sc-4033) were from Santa Cruz. Horseradish peroxidase-conjugated antibody to rabbit IgG (7074) or to mouse IgG (7076) (1:3000 dilution for each) were from Jackson ImmunoResearch Laboratories. The protein bands were visualized with a SuperSignal West Pico Kit according to the manufacturer's instructions (Thermo Fisher Scientific Pierce).Uncropped scans of all western blots are placed in the Supplementary Figs. 5–10.

**In vitro kinase assay**. Mst2 kinase was expressed as a His-tagged protein in E. coli BL21. The Mst2 kinase assay was performed by incubation of His-tagged Mst2, immunoprecipitated with His beads (sigma, P6611), together with recombinant GST–Keap1 amino terminus (amino acids 1–150) or GST–Keap1 amino terminus with the T51A, S53A, T55A, T80A, or combined 4 amino acids mutated with A (4A) substitution (each purified in bacteria) in a kinase assay buffer containing 2 mM dithiothreitol, 10 mM MgCl$_2$, 50 μM ATP, 40 mM HEPES (pH 7.4), and 1 mM EDTA. The reactions were incubated for 30 min at 30 °C before SDS-PAGE and immunoblot analysis.

**CFSE labeling**. BMDMs were labeled for 8 min at 37 °C with 2.5 μM CFSE (carboxyfluorescein diacetate succinimidyl ester) according to manufacturer's instructions (Invitrogen), then were washed three times in complete DMEM, digested with trypsin and mixed with unlabeled cells.

**Confocal fluorescence microscopy**. For paraffin-embedded tissue sections, the spleen were fixed overnight in 10% neutral-buffered formalin and then were dehydrated in increasing concentrations of alcohol, followed by clearing of alcohol with xylene. The specimens were subsequently embedded in paraffin wax in cassettes for facilitation of tissue sectioning. Spleen sections were deparaffinized and incubated in citrate buffer at 95 °C for 40 min for antigen retrieval and then incubated overnight at 4 °C with the primary antibodies including anti-Nrf2 (1:100 dilution, Abcam, ab31163), anti-FoxO1 (1:100 dilution, Cell Signaling Technology, 2880), anti-CD3 (1:500 dilution, Santa Cruz, 137096), and anti-F4/80 (1:100 dilution, Abcam, ab6640). After three washes with PBS, spleen sections were incubated for another 1 h with secondary antibodies (1:250, Alexa Fluor 555-conjugated anti-Rabbit IgG (A31572), Alexa Fluor 488-conjugated anti-Rat IgG (A21470) or Alexa Fluor 488-conjugated anti-mouse IgG (A21202); all from Invitrogen). Subsequently, the sections were washed three times with PBS and were mounted with Vectashield mounting medium containing DAPI (Vector Labs). All images were collected with a confocal microscope (Zeiss LSM 780).

For fluorescence analysis of other molecules in cultured cells, BMDMs seeded on glass coverslips in six-well dishes were stimulated with DMSO, antimycin A (10 μM) or H$_2$O$_2$ for 30 min as needed. HeLa cells seeded on coverslips in six-well dish with 30% confluence were transfected with the appropriate constructs and were cultured for another 24 h. The cells were washed three times with PBS and were fixed for 15 min at room temperature with 4% (vol/vol) paraformaldehyde, after which additional immunofluorescence staining was applied. For staining with p-H2A.X (1:100 dilution, Cell signaling, 9718), anti-Mst1 (1:100 dilution, Proteintech, 22245-1-AP), anti-Tomm20 (1:100 dilution; Abcam, ab56783), anti-Myc tag (1:200 dilution, Santa Cruz, sc-40) or anti-HA tag (1:200 dilution, Santa Cruz, sc-805), fixed cells were rinsed with PBS and then were incubated for 10 min on ice with 0.2% Triton X-100 and 0.2% bovine serum albumin (BSA) in PBS. Following permeabilization, nonspecific binding in the cells was blocked by incubation for 1 h at room temperature with 0.02% Triton X-100 and 5% BSA in PBS and cells were incubated for 1 h with specific primary antibodies (identified above). After three washes with PBS, the cells were incubated for another 1 h with secondary antibodies Alexa Fluor 555-conjugated anti-rabbit IgG (A31572) or Alexa Fluor 488-conjugated anti-mouse IgG (A21202) (both from Invitrogen). Subsequently, the cells were washed three times with PBS and were mounted with Vectashield mounting medium containing DAPI (Vector Labs). All images were collected with a confocal microscope (Zeiss LSM 780) or a Precision DeltaVision-OMX Super-Resolution Microscope (GE OMX V4).

**Protein carbonylation measurement**. Protein carbonyl concentration was determined in BMDMs by using Protein Carbonyl Content Assay Kit (MAK-094, Sigma-Aldrich) according to the manufacturer's instructions. The protein carbonyl assay is based on measuring a stable carbonyl group formed as a result of ROS-oxidized proteins. The amount of protein carbonyl in each sample was reported as nmol protein carbonyl per mg of total protein. The concentration of total protein was determined by using the Bicinchoninic Acid Protein Assay Kit (BCA1, B9643, Sigma-Aldrich) according to the manufacturer's instructions.

**Transmission electron microscopy**. The mouse peritoneal cells were pelleted by centrifugation at 3000 r.p.m. and then primarily fixed in a cold fixing solution containing 2.5% glutaraldehyde for 2 h. The cells mass was sliced to small pieces and fixed in a PBS solution with 2.5% glutaraldehyde (v/v) immediately on ice for

2 h. The primarily fixed samples were further sliced to a volume no larger than 2 mm, rinsed with PBS for three times and then kept in 1% osmium tetraoxide (v/v) for 1 h for the secondary fixation. After dehydration in a series of concentrations of ethanol, the tissues were finally embedded in Embed 812 resin. Thin sections were cut on an ultramicrotome (LKB, Sweden), mounted on nickel grids, stained with saturated uranyl acetate and Reynolds lead citrate, and then examined with electron microscopy JEM2100HC (Jeol Ltd., Japan).

**Telomere length measurement**. Genomic DNA was extracted directly from peritoneal cells using a Mini Genomic DNA Kit (QIAamp DNA Mini Kits) according to manufacturer's instructions (Qiagen). Telomere length was determined by using a real-time quantitative PCR method[64]. The premise of this assay is to measure an average telomere length ratio by quantifying telomeric DNA with specially designed primer sequences and divide that amount by the quantity of a single-copy gene, the acidic ribosomal phosphoprotein PO (36B4) gene, which is well-conserved and has been used for gene-dosage studies. For the telomere portion of the assay, the reaction included 12.5 μl Hieff$^{TM}$ qPCR SYBR® Green Master Mix (Yeasen, 11201ES03), 300 nM each of the forward and reverse primers, 20 ng genomic DNA, and enough double-distilled H$_2$O to yield a 25-μl reaction, and then the reaction was incubated at 95 °C for 10 min followed by 30 cycles of 95 °C for 15 s and a 56 °C anneal–extend step for 1 min in an automated thermo cycler (BIO RAD CFX96, c1000). The reaction for the 36B4 portion contained 12.5 μl Hieff™ qPCR SYBR® Green Master Mix, 300 nM forward primer, 500 nM reverse primer, and enough double-distilled H2O to yield a 25-μl reaction, and then was incubated at 95 °C for 10 min followed by 35 cycles at 95 °C for 15 s, with 52 °C annealing for 20 s, followed by extension at 72 °C for 30 s. Each assay was performed on two 20-ng samples of DNA from each animal, placed in adjacent wells. The primer sequences are listed in Supplemental Table 1.

**Telomere FISH**. Telomere FISH was performed by using a PNA probe (Panagene) described as follows[65,66]. Peritoneal suspensions were collected and were adjusted to $1 \times 10^6$ cells ml$^{-1}$ in DMEM. Then the cells were added to a six-well culture plates with glass slides and incubated at 37 °C for 2 h to allow adherence of macrophages to the glass slides surface. After the incubation, the supernatants containing nonadherent cells (mainly lymphocytes) were removed by gently washing three times with warm PBS. Adhered cells were swollen in KCl buffer (12.3 mM HEPES, 0.53 mM EGTA, 64.4 mM KCl), fixed in methanol/acetic acid (3:1), rehydrated in PBS, fixed in 4% formaldehyde and then dehydrated in a series of concentrations of ethanol. Slides were incubated with hybridization mixture (70% formamide, 10 mM NaHPO$_4$, pH 7.4, 10 mM NaCl, 20 mM Tris buffer, pH 7.5), placed on a 80 °C heating block for 5 min to denature chromosomal DNA and incubated with the PNA probe (0.05 mg ml$^{-1}$) for 2 h at room temperature. After washing, slides were mounted with Vectashield mounting medium containing DAPI (Vector Labs) and analyzed with a confocal microscope (Zeiss LSM 780).

**Recombinant retrovirus and bone marrow transplantation**. Genes encoding Nrf2 were cloned into retroviral vector pMIG containing IRES-regulated GFP. Plat-E packaging cells were transfected with 3 μg of retroviral vector along with 9 μl of TransIT-LT1 transfection reagent (Mirus). Forty-eight hour after transfection, the culture supernatant containing retroviruses was collected. Donor mice (Mst1$^{fl/}$$^{fl}$Mst2$^{fl/fl}$-Lyz Cre) were primed by 5-FU (s1209, 150 mg kg$^{-1}$, selleck,). Seven days later, the BM cells were harvested and cultured with 50 ng ml$^{-1}$ SCF, 10 ng ml$^{-1}$ IL-3 and 20 ng ml$^{-1}$ IL-6 for 12 h. BM cells were then infected with retrovirus together with 8 μg ml$^{-1}$ polybrene by centrifugation of cells at 2000 r.p.m. for 60 min at room temperature. Infected cells were cultured for another 1 or 2 day and ready for bone marrow transplantation. The retroviruses infected donor BM cells ($1.5 \times 10^6$ cells) were intravenously injected into lethally irradiated (9 Gy in one dose) Mst1$^{fl/fl}$Mst2$^{fl/fl}$ recipient mice. After 6 months, mixed chimerism cells were harvested from peritoneal cavity and GFP-positive cells were sorted using flow cytometry for the following analysis. The primer sequences for specific genes are shown in Supplementary Table 1.

**Generation of recombinant adenovirus**. Recombinant adenovirus was generated using standard techniques. Briefly, the Nrf2 or Keap1 cDNA was cloned into the BglII–XhoI restriction sites of the AdTrack-CMV shuttle vector, which contains a GFP as a reporter gene, using the Exonuclease III-LIC systems. The resultant plasmid was linearized by digestion with PmeI endonuclease and cotransformed into E. coli BJ5183 cells with the AdEasy-1 adenoviral plasmid, which contains the entire genomic sequences of adenovirus serotype Ad5, except the nucleotides encompassing the E1 and E3 genes. Recombinant bacteria were selected by kanamycin resistance and the recombination was confirmed by a PacI endonuclease restriction analysis. The control Ad-GFP was constructed similarly. Subsequently, the verified clone was amplified, linearized with PacI and transfected into HEK293A packaging cells. The recombinant adenovirus was released from the cells by four freeze–thaw–vortex cycles 4–7 days post-transfection, amplified by further rounds of infection of HEK293A packaging cells and purified by CsCl ultracentrifugation. Stocks of purified adenovirus were titred by counting the number of plaque-forming units. The recombinant adenoviruses ($2 \times 10^9$ pfu in 200 ml PBS)

were used to infect the Mst1/2-deficient BMDMs. The primer sequences for specific genes are shown in Supplementary Table 1.

**Mass spectrometry**. After staining of gels with Coomassie blue, excised gel segments were subjected to in-gel trypsin digestion and dried. Peptides were dissolved in 10 μl 0.1% formic acid and were auto-sampled directly onto a 100-μm × 10-cm fused silica emitter made in-house and packed with reversed-phase ReproSil-PurC18-AQ resin (3 μm and 120 Å; Ammerbuch). Samples were then eluted for 60 min with linear gradients of 5–32% acetonitrile in 0.1% formic acid at a flow rate of 300 nl min$^{-1}$. Mass spectra data were acquired with a TripleTOF 5600 + mass spectrometer (AB Sciex) equipped with a nano-electrospray ion source. Data were collected at an IDA mode. The wiff files were searched with the ProteinPilot (Version 4.5) against a database from the Uniprot protein sequence database.

**Statistical analysis**. All statistical analyses were performed with Prism6 software (GraphPad Software). Student's t-test was used for comparisons between two groups. A $P$ value of less than 0.05 was considered statistically significant.

**Reporting summary**. Further information on experimental design is available in the Nature Research Reporting Summary linked to this article.

## Data availability

The data that support the findings of this study are available from the corresponding author upon reasonable request.

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

## Acknowledgments

We thank Drs. Luming Yao and Caiming Wu for technical support with the TEM experiments, and Dr. Changchuan Xie for performing mass spectrometry analysis. D.Z. and L.C. were supported by the National Key R&D Program of China (2017YFA0504502 to D.Z. and L.C., 2015CB910502 to L.C.), the National Natural Science Foundation of China (81830046 to L.C.; 31625010, 81790254, and U1505224 to D.Z.; 31600698 to J. Geng.; 81871305 to X.L.; 3170070 to X.S.), the Fundamental Research Funds for the Central Universities of China-Xiamen University (20720180047 to L.C.), China Post-doctoral Science Foundation (2016M602072 and 2017T100470 to J. Geng), Young Elite Scientist Sponsorship Program by CAST (2017QNRC001 to J. Geng), the 111 Projects (BC2018027, B12001 and BP2018017). The funders had no role in study design, data collection and analysis, decision to publish, or preparation of the manuscript.

## Author contributions

D.Z. and L.C. conceived the project with input from J.S.Y., Z.Y., H.Y. and R.L.J. P.W., J. Geng, J. Gao, H.Z., J.L., Y.S., C.X., B.Y., Y.L., X.S., L.H, F.Q. and X.L. performed experimental biological research. X.C. performed super-resolution immunofluorescence microscopy analysis. R.L.J. provided mutant mice. L.C. and D.Z. co-wrote the paper. All authors edited the manuscript.

## Additional information

**Competing interests:** The authors declare no competing interests.

