## [Peer Review File · Nature Communications]

Reviewers' comments:

Reviewer #1 (Immune ageing, senescence)(Remarks to the Author):

This is a very detailed characterisation of the role of mst1 in macrophages and provides a comprehensive analysis of its mechanism of action. My problem with the paper comes in the definition of its role in ageing. The paper has artificially made the cells senescent by giving/making the cells produce more ROS. It has been well established that ROS causes telomere shortening but this is not ageing – especially when most immune cells, innate and adaptive undergo telomere-independent as well as telomere-dependent senescence. It would have been more convincing if the authors could show that mst1 is reduced in old animals or cultured macrophages. This together with something functionally relevant to aged macrophages, ie loss of function, decreased respiratory burst, inability to clear bacteria. As it is I'm not convinced that the paper represents a potential new mechanism that controls ROS in aged macrophages, also the free-radical theory of ageing has largely been discredited owing to a lack of evidence, therefore the central tenet of the paper is largely flawed. Minor point not enough detail has been given about the PCR telomerase assay.

Reviewer #2 (Inflammation, M1/M2 macrophage)(Remarks to the Author):

This manuscript describes a role for Mst1/2 in sensing oxidative stress and mediating anti-oxidant defense through the Keap1-Nrf2 axis. This was a follow up study of authors' previous publication, in which they showed myeloid Mst1/2 knockout (cDKO) macrophages have defective ROS production, as Mst1/2 are required for mitochondrion-phagosome juxtaposition (Nature Immunology 16:1142). Here, the authors further demonstrated that in the absence of Mst1/2, Keap1-mediated Nrf2 degradation was enhanced leading to a reduction in anti-oxidant gene expression and an increase in oxidative damage. The authors proposed that Keap1 phosphorylation by Mst1/2 disassociates Keap1/Nrf2, allowing stabilization and nuclear translocation of Nrf2 to control anti-oxidant gene expression.

Although the proposed mechanism is interesting, the authors left out important information that cDKO macrophages fail to produce ROS in response to certain TLR ligands, which could explain the less stabilized Nrf2 protein under stimulation. Also, if Mst1/2 are sequestered to the mitochondria/phagosome by infection/TLR activation, how does the kinase complex phosphorylate cytosolic Keap1/Nrf2? Additional experiments are needed to clarify these concerns.

Specific comments

1. Data presentation is a bit misleading. Fig. 1 and 2 focus on basal ROS, which is moderately elevated in cDKO (which was how the authors described in the original study in Nature Immunology), as well as oxidative damage presumably associated with it. However, Nrf2 activity is typically very low under unstimulated condition. The original study also pointed out that the difference was not in ROS produced by mitochondria, but most experiments done in the current study used mitochondria complex inhibitors to trigger mROS. The relevance of the experimental approach is questionable and should be justified.
2. Along the same line, the authors should perform additional experiments using relevant stimulants, such as GFP-E. coli and TLR ligands, in WT and cDKO macrophages for comparison. Fig. 4A and 6J need to include gene/protein expression under stimulations (e.g., TLR ligands).
3. In Fig. 2e, it's unclear why Mst1 showed a punctate staining pattern. What's the cellular localization? The authors should include Keap1 (and Nrf2) staining at basal and stimulated states to determine when and where Mst1/2 interact with Keap1. Since most of the biochemical interaction characterizations used transient over-expression, it's important to provide additional validations with

endogenous proteins.

4. The Nrf2 over-expression result in Fig. 7b does not really show Nrf2 over-expression rescued the phenotype, as the WT +/- AdGFP or AdNrf2 controls were missing. A better experiment would be to use WT and mutant Keap1 described in Fig. 6J +/- stimulation in WT and cDKO macrophages.

Minor comments:

1. In Fig. 3d-3g, b-actin and HSP60 were used to demonstrate the purity of fractionation. A small amount of HSP60 is known to be in the cytosol and actin is involved in the mitochondrial fission process. It's unclear how the authors obtained such a clear/clean result. Other mitochondrial and cytosol markers should be included.

2. Immune cell aging or immunosenescence has been defined (although still not completely understood). Fig. 1 and 2 are descriptive results showing oxidative damage. The authors should avoid using the term unless more functional characterization will be performed.

Reply to Reviewer #1:

We would like to thank the reviewer for the positive comments on our manuscript. In response to these insightful comments, we now have revised manuscript with additional new data. Please note that all changes in the revised manuscript are highlighted with yellow.

Reviewer #1 (Immune ageing, senescence)(Remarks to the Author):

This is a very detailed characterisation of the role of mst1 in macrophages and provides a comprehensive analysis of its mechanism of action. My problem with the paper comes in the definition of its role in ageing. The paper has artificially made the cells senescent by giving/making the cells produce more ROS. It has been well established that ROS causes telomere shortening but this is not ageing – especially when most immune cells, innate and adaptive undergo telomere-independent as well as telomere-dependent senescence. It would have been more convincing if the authors could show that mst1 is reduced in old animals or cultured macrophages. This together with something functionally relevant to aged macrophages, ie loss of function, decreased respiratory burst, inability to clear bacteria. As it is I'm not convinced that the paper represents a potential new mechanism that controls ROS in aged macrophages, also the free-radical theory of ageing has largely been discredited owing to a lack of evidence, therefore the central tenet of the paper is largely flawed. Minor point not enough detail has been given about the PCR telomerase assay.

Answer: We apologize to the reviewer that we did not explain the conclusion in a clear manner. Recently, Meng J et al reported a new concept that the decay of Redox-stress Response Capacity (RRC) is a substantive characteristic of aging, which revises the redox theory of aging. RRC refers to the ability of cells to respond to oxidative stress, specifically three major activities: the ability to generate ROS or RNS, the ability to regulate antioxidants, and the ability to degrade damaged proteins for maintaining cellular redox and protein homeostasis (Meng J et al. Redox biology. 2017; 11: 365-374). We have previously reported that Mst1/2 null macrophage has defect in generation of ROS upon bacterial infection (Geng J et al. Nat Immunol. 2015 Nov;16(11):1142-52.). In current manuscript, we also found that Mst1/2 deficient macrophages undergo chronic oxidative stress and have reduced RRC ability, indicating cells undergoing premature ageing. With comprehensive mechanism study, we revealed that Mst1/2 kinases protect cells against oxidative damage and senescent by regulating the function of Keap1-Nrf2.

We agree with the reviewer's concern that telomere shortening is not sufficient to mark macrophage senescence. Following the reviewer's constructive suggestion, we determined the expression levels of Mst1, Mst2, as well as p-Mob1, the substrate of Mst1/2, in peritoneal macrophages isolated from mice at different age. Indeed, we observed that the mRNA and protein levels of Mst1 and Mst2 were significantly reduced in aged macrophages (20 months)

(Figure R1A and B). It has been previously shown that activated peritoneal macrophages from aged mice expressed significantly lower levels of TLRs (1-9) (Renshaw M et al, J Immunol. 2002;169(9):4697-701.). Indeed, our analysis confirmed that the expression levels of various TLRs (1-9) were much lower in aged wild type macrophages (20 months) when compared with that of macrophages isolated from younger mice (2 or 12 months). In comparison of the TLRs in WT and DKO macrophages, we found that the expression levels of TLR1, 2, 5, 8 and 9 were not significant difference in 2-month old macrophages, but were significantly decreased in 12- and 20-month DKO macrophages. Whereas, the expression levels TLR3, 4, 6 and 7 were all much lower in DKO macrophages compared with that of WT macrophages at any given ages (Figure R1C). Regarding the functionally relevant to aged macrophages, we have previously reported that DKO macrophage had decreased respiratory burst and inability to clear bacteria (Figure R1D and E) (Geng J et al. Nat Immunol. 2015 Nov;16(11):1142-52.). In additions, we observed that the ability of macrophages to activate OVA-specific OT-II CD4⁺ T cells as measured by the proliferation assay was gradually decreased with increasing age (Figure R1F). Consistently, OT-II CD4⁺ T cells exhibit much lower proliferation rate when incubated with 12- or 20-month DKO macrophages when compared with that of the same ages WT macrophages. Taken together, we conclude that Mst1/2-deficient macrophages undergo premature ageing or senescence as marked with decreased ability to generate ROS and regulate antioxidants, as well as defects on immune functions similar to aged macrophages, such as TLRs expression, bacteria clearance and antigen presentation for activation of CD4⁺ T cells. We have added these data in the revised manuscript (Fig. 2a, b, d and e in the revised manuscript).

Figure R1

Fig. R1. (A and B) RT-qPCR of *Mst1* and *Mst2* (A), and immunoblot analysis of *Mst1*, *Mst2* and p-Mob (B) in macrophages isolated from WT mice with indicated age. (C) The expression levels of various TLRs on peritoneal macrophages isolated from WT or DKO mice with indicated age. (D) Flow cytometry analyzing the production of mROS and cellular ROS in wild-type (WT) and *Mst1/2* DKO BMDMs infected *E. coli* or *L. monocytogenes* (*Lm*) and then stained for 30 min with MitoSOX or CM-H₂DCFDA. (E) Pathogen burden in WT and DKO BMDMs infected with *Lm* (MOI, 10) or *E. coli* (MOI, 20), presented as colony-forming units (CFU). (F) Proliferation assay of CFSE-labeled CD4⁺ OT-II T cells co-cultured with WT and cDKO BMDMs in the presence of OVA protein.

Reply to Reviewer #2:

We would like to thank the reviewer for the positive comments on our manuscript. In response to these insightful comments, we now have revised manuscript with additional new data. Please note that all changes in the revised manuscript are highlighted with yellow.

Reviewer #2 (Inflammation, M1/M2 macrophage) (Remarks to the Author):

This manuscript describes a role for Mst1/2 in sensing oxidative stress and mediating anti-oxidant defense through the Keap1-Nrf2 axis. This was a follow up study of authors' previous publication, in which they showed myeloid Mst1/2 knockout (cDKO) macrophages have defective ROS production, as Mst1/2 are required for mitochondrion-phagosome juxtaposition (Nature Immunology 16:1142). Here, the authors further demonstrated that in the absence of Mst1/2, Keap1-mediated Nrf2 degradation was enhanced leading to a reduction in anti-oxidant gene expression and an increase in oxidative damage. The authors proposed that Keap1 phosphorylation by Mst1/2 disassociates Keap1/Nrf2, allowing stabilization and nuclear translocation of Nrf2 to control anti-oxidant gene expression.

Although the proposed mechanism is interesting, the authors left out important information that cDKO macrophages fail to produce ROS in response to certain TLR ligands, which could explain the less stabilized Nrf2 protein under stimulation.

Answer: We agree with the reviewer's concern and apologize that we did not make the presentation clear. Our previous study showed that TLR-stimulated cDKO macrophages did not generate as much total and mitochondrial ROS as that in WT macrophages since Mst1/2 act downstream of TLR signaling to trigger juxtaposition of the mitochondria and phagosome to augment ROS production (Geng J et al. Nat Immunol. 2015 Nov;16(11):1142-52.). In contrast, we found that rotenone, oligomycin and antimycin A, which are electron-transport inhibitors known to increase mitochondrial generation of superoxide, augmented mROS production to a similar extent in wild-type and cDKO BMDMs. In addition, the number of mitochondria, the main electron-transport-chain components and the mitochondrial membrane potential were also similar in wild-type and cDKO cells (Geng J et al. Nat Immunol. 2015 Nov;16(11):1142-52.). Thus, a lack of Mst1 and Mst2 did not diminish the capacity of mitochondria for ROS production. To compare the antioxidant capability in WT and cDKO macrophages, we treated macrophage with antimycin A or rotenone, which produces similar levels of oxidative stress in WT and cDKO macrophages. We observed that less stabilized Nrf2 protein in DKO macrophages than that of WT macrophages upon the treatment of antimycin A or rotenone. These data suggest that Mst1/2 kinases are required for Nrf2 stability.

Also, if Mst1/2 are sequestered to the mitochondria/phagosome by infection/TLR activation, how does the kinase complex phosphorylate cytosolic Keap1/Nrf2? Additional experiments are needed to clarify these concerns.

Answer: Thanks a lot for the reviewer's insightful comment. We observed that Keap1 was recruited with Mst1/2 from the cytosol to the mitochondrial membrane when BMDMs were treated with *E. coli* or antimycin A (Figure R2A-B). With super-resolution immunofluorescence microscopy (SIM) approach, we observed Mst1 protein condensed and formed large aggregates after *E. coli* infection (from 5 min and at least last for 60 min). Keap1 co-localized with Mst1 after *E. coli* infection at 5 /15/ 30 min time points, but not at 60 min. Previous study has shown that, under unstressed conditions, Keap1 forms dimer/polymer to promote Nrf2 degradation through the ubiquitination-mediated proteasome pathway (Mol. Cell Biol. 2004; 24: 7130-7139). In current manuscript we found that Mst1/2-mediated Keap1 phosphorylation disrupts Keap1 dimerization or polymerization. Consistently, with SIM approach, Keap1 (green) exhibited strong localization with Mst1, bright aggregates in BMDMs without infection or upon *E. coli* infection for the first 30 min, and at 60 min post-infection Keap1 became much less condensed with Mst1 and more diffused than that in BMDMs without infection (Figure R2C). We suspect that, at the beginning of infection (5 min to 30 min), Mst1/2 kinases interact and phosphorylate Keap1 leading to Keap1 de-polymerization, thus the less aggregation of Keap1 was observed at 60 min post- *E. coli* infection.

Figure R2

C

Fig. R2. (A and B) Immunoblot analysis of Mst1, Mst2, β-actin and VDAC in the cytoplasmic (Cyto) and mitochondrial (Mito) fractions of WT BMDMs pretreated with or without NAC and infected with *E. coli* (MOI: 100) (A) or treated with antimycin A (B) for the indicated time. (C) SIM of the co-localization of Mst1 (red) and Keap1 (green) in WT BMDMs infected with *E. coli* (blue) for the indicated time; 16× magnification of areas outlined in the main images are shown next to the main images. Scale bars, 20 μm.

Specific comments

1. Data presentation is a bit misleading. Fig. 1 and 2 focus on basal ROS, which is moderately elevated in cDKO (which was how the authors described in the original study in Nature Immunology), as well as oxidative damage presumably associated with it. However, Nrf2 activity is typically very low under unstimulated condition.

Answer: Yes, we agree with the reviewer that Nrf2 is typically very low under unstimulated condition. However, we analyzed Nrf2 knockout mice and found that loss of Nrf2 also resulted in elevated basal ROS in macrophages without any stimulation (Figure R3), suggesting the basal Nrf2, although at very low basal level, is essential for maintaining cellular basal antioxidative defense under unstimulated condition.

Figure R3 (updated in Fig. 4f)

Fig. R3. Flow cytometry analysis of ROS levels of Nrf2-deficient BMDMs and their WT counterparts. The fluorescence intensity of the ROS dye in the indicated cell samples is shown in the right panel.

The original study also pointed out that the difference was not in ROS produced by mitochondria, but most experiments done in the current study used mitochondria complex inhibitors to trigger mROS. The relevance of the experimental approach is questionable and should be justified.

Answer: We apologize to the reviewer that we did not make the description clear. In current manuscript, we observed that the increased basal ROS level and enhanced oxidative stress in cDKO macrophage, which suggested that the Mst1/2 deficient macrophage might have a defect in sensing and clearance of ROS.

Our previous study showed that TLR-stimulated cDKO macrophages did not generate as much total and mitochondrial ROS as that in WT macrophages since Mst1/2 act downstream of TLR signaling to trigger juxtaposition of the mitochondria and phagosome to augment ROS production (Geng J et al. Nat Immunol. 2015 Nov;16(11):1142-52.). Thus, TLR stimulation could not provide a fair condition to compare the antioxidant response between WT and cDKO macrophages. In contrast, we found that rotenone, oligomycin and antimycin A, which are electron-transport inhibitors known to increase mitochondrial generation of superoxide, augmented mROS production to a similar extent in wild-type and cDKO BMDMs. In addition, the number of mitochondria, the main electron-transport-chain components and the mitochondrial membrane potential were also similar in wild-type and cDKO cells (Geng J et al. Nat Immunol. 2015 Nov;16(11):1142-52.). Thus, a lack of Mst1 and Mst2 did not diminish the capacity of mitochondria for ROS production. Moreover, we found that the cellular ROS levels were also increased by the treatment of antimycin A and rotenone, which provides a tool to study the antioxidant response in Mst1/2 deficient macrophages (Figure R4). Thus, in current manuscript, to compare the antioxidant capability in WT and cDKO macrophages during oxidative stress, we took the advantage of antimycin A or rotenone to

induce equal amount of mROS/cellular ROS for triggering the antioxidant response.

Figure R4 (updated in Supplemental Fig. 1d)

Fig. R4. Flow cytometry analysis of mROS (MitoSOX) and total cellular ROS (CellROX) levels of WT and cDKO BMDMs treated with antimycin A (10 μ M) or rotenone (3 μ M).

2. Along the same line, the authors should perform additional experiments using relevant stimulants, such as GFP-*E. coli* and TLR ligands, in WT and cDKO macrophages for comparison. Fig. 4A and 6J need to include gene/protein expression under stimulations (e.g., TLR ligands).

Answer: Following the reviewer's constructive suggestion, we treated WT or cDKO macrophages with LPS, *E. coli*, as well as antimycin A and rotenone. The results showed that the expression levels of various antioxidant enzymes of WT macrophages were significantly higher than that of DKO macrophages upon the stimulations (Figure R5A). In additions, we found that overexpressing non-phospho mutant Keap14A, which mimics Mst1/2 deficiency, but not phospho-mimic mutant Keap14D profoundly downregulated the expression levels of Nrf2, NQO1 and HO-1 in Raw264.7 cells upon *E. coli* infection or H₂O₂ treatment (Figures R5B-D).

Figure R5

A (updated in Fig. 4a)

B (updated in Fig. 4b-d)

C (updated in Fig. 6k)

D (updated Fig. 6j)

E

Fig. R5. (A) RT-qPCR analysis of the mRNA levels of the indicated antioxidant genes in WT and cDKO BMDMs infected with *E. coli* (MOI: 100) or treated with LPS, antimycin A or rotenone. (B) Immunoblot analysis of Gclc, Gclm, Nqo-1, Ho-1, p-Mob1, Mob1, Mst1, Mst2 and GAPDH in WT and cDKO BMDMs infected with *E. coli* or treated with LPS or H₂O₂. (C) RT-qPCR analysis of the mRNA levels of the antioxidant genes *Nqo1* and *Ho-1* in Raw264.7 cells expressing HA-tagged Keap1^{WT}, Keap1^{4A} or Keap1^{4D} followed *E. coli* infection for 3 h. (D and E) Immunoblot analysis of NQO1, HO-1, Nrf2, Keap1 and GAPDH in Raw264.7 cells infected with *E. coli* or treated with H₂O₂ for 3 h.

3. In Fig. 3e, it's unclear why Mst1 showed a punctate staining pattern. What's the cellular localization?

Answer: We used DeltaVision OMX super-resolution microscope approach, which is an advanced instruments based on SI (Structured Illumination, SI or SIM) technology, to visualize the co-localization of Mst1 with the ROS generating compartments. SIM imaging approximately doubles the resolution both axially and laterally compared with conventional wide-field microscopy (Gustafsson MG, J Microsc. 2000 May;198(Pt 2):82-7.). The general principle is that a grid pattern can be superimposed upon another grid pattern at different angles to yield a third pattern containing Moiré fringes. These three patterns are mathematically related such that if the spatial frequencies of two of the three patterns are

known, the frequency of the third can be back calculated. Thus, in SIM, light is structured upon a sample (pattern 1) in a grid pattern (pattern 2), and the resulting Moiré fringe (pattern 3) is collected via the objective in different planes (MacDonald L et al. *Methods Mol Biol.* 2015; 1270:255-75. doi: 10.1007/978-1-4939-2309-0_19.). As the following figure showed SIM technology increases resolution in the XY dimension and increases contrast over conventional techniques.

A single image plane from a human platelet viewed by wide field microscopy (A), after deconvolution (B), and by 3D-SIM microscopy (C). (Adapted from the Figure 1 in *Methods Mol Biol.* 2015; 1270: 255-75)

Following the reviewer's suggestion, we compared the staining pattern of Mst1 in BMDMs using the conventional confocal microscope and the SIM approach. We found that the confocal image showed a much less clear punctate staining pattern as compared with the SIM approach. Notably, the punctate staining pattern of Mst1 was not present in Mst1 deficient BMDMs, indicating the punctate staining pattern of Mst1 in BMDMs by SIM is real. In additions, our result showed that, in general, majority of Mst1 is localized in cytoplasm (Figure R6). In the future, we will determine whether Mst1 is specially co-localized with certain kinds of cellular organelle.

Figure R6

Fig. R6. (A) Fluorescence microscopy of Mst1 staining (red) and DAPI-stained nuclei (blue) in CFSE-labeled WT BMDMs (the upper panel) or DKO DMDMs (the lower panel) mixed with unlabelled DKO BMDMs (the upper panel) or WT BMDMs (the lower panel) respectively. Scale bars, 20 μ m. (B) SIM of Mst1 staining (red) in WT BMDMs (the upper panel) or cDKO DMDMs (the lower panel); 16 \times magnification of areas outlined in the main images are shown next to the main images. Scale bars, 20 μ m.

The authors should include Keap1 (and Nrf2) staining at basal and stimulated states to

determine when and where Mst1/2 interact with Keap1. Since most of the biochemical interaction characterizations used transient over-expression, it's important to provide additional validations with endogenous proteins.

Answer: We agree with the reviewer's concerns. Since there are not suitable immunoprecipitation (IP) antibodies for endogenous Mst1 or Mst2, we used the immortalized BMDMs cell line stably expressing Flag-tagged Mst2, relatively close to endogenous expression levels generated previously in our lab to determine the interaction of Keap1 and Mst2 in macrophages. We found that the interaction of Keap1 and Mst2 was enhanced upon *E. coli* infection or antimycin A (AA) treatment started from 10 min and lasted at least for 30 min (Figure R7A). In addition, we also observed that Keap1 was recruited with Mst1/2 from the cytosol to the mitochondrial membrane when WT BMDMs were treated with *E. coli* or antimycin A (Figures R7B and C). With super-resolution immunofluorescence microscopy (SIM) approach, we observed Mst1 protein condensed and formed large aggregates after *E. coli* infection (from 5 min and at least last for 60 min). Keap1 co-localized with Mst1 after *E. coli* infection at 5 /15/ 30 min time points, but not at 60 min. Previous study has shown that, under unstressed conditions, Keap1 forms dimer/polymer to promote Nrf2 degradation through the ubiquitination-mediated proteasome pathway (Mol. Cell Biol. 2004; 24: 7130-7139). In current manuscript we found that Mst1/2-mediated Keap1 phosphorylation disrupts Keap1 dimerization or polymerization. Consistently, with SIM approach, Keap1 (green) exhibited strong localization with Mst1, bright aggregates in BMDMs without infection or upon *E. coli* infection for the first 30 min, and at 60 min post-infection Keap1 became much less condensed with Mst1 and more diffused than that in BMDMs without infection (Figure R2C). We suspect that, at the beginning of infection (5 min to 30 min), Mst1/2 kinases interact and phosphorylate Keap1 leading to Keap1 de-polymerization, thus the less aggregation of Keap1 was observed at 60 min post- *E. coli* infection.

Figure R7

A (updated in Fig. 5e and Supplementary Fig. 2d).

B (updated in Fig. 5f)

C (updated in Supplementary Fig. 2e)

D

Fig. R7. (A) Immunoblot analysis of Keap1 and Mst2 in anti-IgG or anti-Flag IP (top) and total lysates (bottom) of WT BMDM cell line stably expressing Flag-tagged Mst2 infected with *E. coli* (MOI: 100) or treated with antimycin A for indicated times. (B and C) Immunoblot analysis of Mst1, Mst2, β -actin and VDAC in the cytoplasmic (Cyto) and mitochondrial (Mito) fractions of WT BMDMs pretreated with or without NAC and infected with *E. coli* (MOI: 100) (B) or treated with antimycin A (C) for the indicated time. (D) SIM of the co-localization of Mst1 (red) and Keap1 (green) in WT BMDMs infected with *E. coli* (blue) for the indicated time; 16 \times magnification of areas outlined in the main images are shown next to the main images. Scale bars, 20 μ m.

4. The Nrf2 over-expression result in Fig. 7b does not really show Nrf2 over-expression rescued the phenotype, as the WT +/- AdGFP or AdNrf2 controls were missing. A better experiment would be to use WT and mutant Keap1 described in Fig. 6J +/- stimulation in WT and cDKO macrophages.

Answer: Following the reviewer's constructive suggestion, both WT and DKO BMDMs were infected with control adenovirus expressing GFP (Ad-GFP) or adenovirus expressing Nrf2 (Ad-Nrf2). Indeed, reintroduction of Ad-Nrf2, but not of Ad-GFP, significantly increased the transcription of the Ho-1 and Nqo1 genes in DKO BMDMs, suggesting Nrf2 is responsible for the anti-oxidant response defects in DKO macrophages (Figures R8A). In additions, we found that overexpressing non-phospho mutant Keap1^{4A}, which mimics Mst1/2 deficiency, but not phospho-mimic mutant Keap1^{4D} profoundly downregulated the expression levels of Nqo1 and Ho-1 in DKO BMDMs upon H₂O₂ treatment (Figure R8B).

Figure R8

A (updated in Fig. 7b)

B

Fig. R8. (A) RT-qPCR analysis of the mRNA levels of the antioxidant genes *Nqo1* and *Ho-1* in WT BMDMs and DKO BMDMs infected with Ad-GFP or Ad-Nrf2, followed by H₂O₂ treatment for 3 h. (B) RT-qPCR analysis of the mRNA levels of the antioxidant genes *Nqo1* and *Ho-1* in WT BMDMs and DKO BMDMs infected with Ad-GFP, Ad-Keap1^{WT}, Ad-Keap1^{4A} or Ad-Keap1^{4D}, followed by H₂O₂ treatment for 3 h.

Minor comments:

1. In Fig. 3d-3g, b-actin and HSP60 were used to demonstrate the purity of fractionation. A small amount of HSP60 is known to be in the cytosol and actin is involved in the mitochondrial fission process. It's unclear how the authors obtained such a clear/clean result.

Other mitochondrial and cytosol markers should be included.

Answer: We rechecked the raw data of corresponding blots in Fig. 3d-3g and found that some longer exposure such as mitochondrial marker Hsp60 blotting did produce some contamination signals in the cytosol fraction (fig.3d). Since some of them lacking long time exposure blotting, we performed another sets experiments (checking the co-localization of Keap1 and Mst1/2 in Figure R2 (A, B) or 7 (B,C)) of mitochondrial and cytosol fractionations. We do see the trace amount of β -actin in mitochondrial fraction and Hsp60 or VDAC (Voltage-dependent anion channel, another mitochondrial marker) in the cytosol fraction with long exposure time of blots (Fig.5f and supplementary Fig.2e).

Fig. R6. Immunoblot analysis of Hsp60 and β -actin in the cytoplasmic (Cyto) and mitochondrial (Mito) fractions of WT BMDMs of the indicated figures. (RAW data)

2. Immune cell aging or immunosenescence has been defined (although still not completely understood). Fig. 1 and 2 are descriptive results showing oxidative damage. The authors should avoid using the term unless more functional characterization will be performed.

Answer: We thank the reviewer for the constructive comments and suggestions. Recently, Meng J et al reported a new concept that the decay of Redox-stress Response Capacity (RRC) is a substantive characteristic of aging, which revises the redox theory of aging. RRC refers to the ability of cells to respond to oxidative stress, specifically three major activities: the ability to generate ROS or RNS, the ability to regulate antioxidants, and the ability to degrade damaged proteins for maintaining cellular redox and protein homeostasis (Meng J et al. Redox biology. 2017; 11: 365-374). We have previously reported that Mst1/2 null macrophage has defect in generation of ROS upon bacterial infection (Geng J et al. Nat Immunol. 2015 Nov;16(11):1142-52.). In current manuscript, we also found that Mst1/2 deficient macrophages undergo chronic oxidative stress and have reduced RRC ability, indicating cells undergoing premature ageing. With comprehensive mechanism study, we revealed that Mst1/2 kinases protect cells against oxidative damage and senescent by regulating the function of Keap1-Nrf2.

We agree with the reviewer's concern that descriptive results showing oxidative damage and telomere shortening is not sufficient to mark macrophage senescence. It has been previously shown that activated peritoneal macrophages from aged mice expressed significantly lower levels of TLRs (1-9) (Renshaw M et al, J Immunol. 2002;169(9):4697-701.). Indeed, our analysis confirmed that the expression levels of various TLRs (1-9) were much lower in aged wild type macrophages (20 months) when compared with that of macrophages isolated from younger mice (2 or 12 months). In comparison of the TLRs in WT and DKO macrophages, we found that the expression levels of TLR1, 2, 5, 8 and 9 were not significant difference in 2-month old macrophages, but were significantly decreased in 12- and 20-month DKO macrophages. Whereas, the expression levels TLR3, 4, 6 and 7 were all much lower in DKO macrophages compared with that of WT macrophages at any given ages (Figure R10A). Regarding the functionally relevant to aged macrophages, we have previously reported that DKO macrophage had decreased respiratory burst and inability to clear bacteria (Figure R10B and C) (Geng J et al. Nat Immunol. 2015 Nov;16(11):1142-52.). In additions, we observed that the ability of macrophages to activate OVA-specific OT-II CD4⁺ T cells as measured by the proliferation assay was gradually decreased with increasing age (Figure R10D). Consistently, OT-II CD4⁺ T cells exhibit much lower proliferation rate when incubated with 12- or 20-month DKO macrophages when compared with that of the same ages WT macrophages. Taken together, we conclude that Mst1/2-deficient macrophages undergo premature ageing or senescence as marked with decreased ability to generate ROS and regulate antioxidants, as well as defects on immune functions similar to aged macrophages, such as TLRs expression, bacteria clearance and antigen presentation for activation of CD4⁺ T cells. In addition, we determined the expression levels of Mst1, Mst2, as well as p-Mob1, the substrate of Mst1/2, in peritoneal macrophages isolated form mice at difference age. Furthermore, we observed that the mRNA and protein levels of Mst1 and Mst2 were significantly reduced in aged macrophages (20 months) (Figure R10E and F) suggesting that Mst1/2 might play an essential anti-ageing role in macrophages. We have added these data in

the revised manuscript (Fig. 2a, b, d and e in the revised manuscript).

Figure R10

A (updated in Fig. 2a)

B (Figure 2h of Geng J et al. NI 2015)

C (Figure 2b of Geng J et al. NI 2015)

D (updated in Fig. 2b)

E (updated in Fig. 2d)

F (updated in Fig. 2e)

Fig. R10. (A) The expression levels of various TLRs on peritoneal macrophages isolated from WT or DKO mice with indicated age. (B) Flow cytometry analyzing the production of mROS and cellular ROS in wild-type (WT) and Mst1/2 DKO BMDMs infected *E. coli* or *L. monocytogenes* (*Lm*) and then stained for 30 min with MitoSOX or CM-H₂DCFDA. (C) Pathogen burden in WT and DKO BMDMs infected with *Lm* (MOI, 10) or *E. coli* (MOI, 20), presented as colony-forming units (CFU). (D) Proliferation assay of CFSE-labeled CD4⁺ OT-II T cells co-cultured with WT and cDKO BMDMs in the presence of OVA protein. (E and F) RT-qPCR of Mst1 and Mst2 (E), and immunoblot analysis of Mst1, Mst2 and p-Mob (F) in macrophages isolated from WT mice with indicated age.

REVIEWERS' COMMENTS:

Reviewer #1 (Remarks to the Author):

Thank you for addressing my concerns. I was unaware of your interesting publication about the RRC and ageing, thank you for informing me.
I believe you've adequately answered all my questions.

Reviewer #2 (Remarks to the Author):

authors have addressed most of my concerns. One minor point is that in Fig. R7D, which was meant to demonstrate interaction between endogenous Mist1 and Keap1, NAC treatment should be include to complete the panels. The data should also be included in Fig. 5.

Reviewer #1 (Remarks to the Author):

Thank you for addressing my concerns. I was unaware of your interesting publication about the RRC and ageing, thank you for informing me.

I believe you've adequately answered all my questions.

Answer: We would like to thank the reviewer for the positive comments on our manuscript.

Reviewer #2 (Remarks to the Author):

authors have addressed most of my concerns. One minor point is that in Fig. R7D, which was meant to demonstrate interaction between endogenous Mist1 and Keap1, NAC treatment should be include to complete the panels. The data should also be included in Fig. 5.

*Answer: We would like to thank the reviewer for the positive comments on our manuscript. We have included the Fig. R7D in the Supplementary Fig. 3f. We have also performed additional experiments to demonstrate interaction between endogenous Mist1 and Keap1 in BMDMs upon *E. coli* infection with or without NAC treatment, and added new data in the updated Fig. 5g.*